# Prospects of Halide Perovskites for Solar-to-Hydrogen Production

**DOI:** 10.3390/nano14231914

**Published:** 2024-11-28

**Authors:** Huilong Liu, Tulja Bhavani Korukonda, Shubhra Bansal

**Affiliations:** 1School of Mechanical Engineering, Purdue University, West Lafayette, IN 47907, USA; 2School of Materials Engineering, Purdue University, West Lafayette, IN 47907, USA

**Keywords:** halide perovskite, solar to hydrogen, efficiency, photocatalysis, photo electrochemical

## Abstract

Solar-driven hydrogen generation is one of the promising technologies developed to address the world’s growing energy demand in an sustainable way. While, for hydrogen generation (otherwise water splitting), photocatalytic, photoelectrochemical, and PV-integrated water splitting systems employing conventional semiconductor oxides materials and their electrodes have been under investigation for over a decade, lead (Pb)- halide perovskites (HPs) made their debut in 2016. Since then, the exceptional characteristics of these materials, such as their tunable optoelectronic properties, ease of processing, high absorption coefficients, and long diffusion lengths, have positioned them as a highly promising material for solar-driven water splitting. Like in solar photovoltaics, a solar-driven water splitting field is also dominated by Pb-HPs with ongoing efforts to improve material stability and hydrogen evolution/generation rate (HER). Despite this, with the unveiling potential of various Pb-free HP compositions in photovoltaics and optoelectronics researchers were inspired to explore the potential of these materials in water splitting. In this current review, we outlined the fundamentals of water splitting, provided a summary of Pb HPs in this field, and the associated issues are presented. Subsequently, Pb-free HP compositions and strategies employed for improving the photocatalytic and/or electrochemical activity of the material are discussed in detail. Finally, this review presents existing issues and the future potential of lead-free HPs, which show potential for enhancing productivity of solar-to-hydrogen conversion technologies.

## 1. Introduction 

Photocatalytic water splitting is one of the efficient green energy technologies which can produce hydrogen fuel with little to no toxic emissions. Ever since the first demonstration of photoelectrochemical water splitting by Honda and Fujishima, solar energy-driven hydrogen generation, employing semiconductors as an active material, created a buzz in the research community [1]. Several materials, including TiO_2_, ZrO_2_, hetero-phase-mixed oxides, other white titanates and zirconates, and oxide perovskite semiconductors have been under constant investigation for hydrogen generation; however, most of these semiconductors are active in the UV region and there had been an ongoing demand to investigate materials with absorption in the visible region or competent with a broader absorption spectrum to utilize a substantial amount of solar radiation. While doping of the existing active materials to tune the bandgap and achieve absorption in the visible region is one approach, alternatively, the materials which are already proven to be satisfactory for other photovoltaic applications (like LEDs and solar cells) intrigued the researchers.

The application of traditional perovskite ABO_3_ as a catalyst for solar hydrogen production is well reviewed in recent articles [2,3]. Materials with AMX_3_ (A being the large cation, M being the small metal cation, and X being halide anion) are often referred to as Halide-perovskites (HPs), while the term perovskite pertains to the material’s crystal structure. These materials are composed of two cations: A is the larger cation, B is the smaller cation, usually a metal ion, and X is the halide anion. The investigation of Pb-HP CsPbX_3_ is dated back to 1958 by Moller [4]. In the following decade, Pb-HPs fascinated researchers with their intriguing optoelectronic and structural properties. Pb-HPs are known for their high light absorption coefficient and high carrier mobility with low excitonic binding energy, which make them potential candidates for photon-induced activities. In 2009, with Miyasaka’s report of replacing dye with organic–inorganic halide perovskite CH_3_NH_3_PbI_3_ as a sensitizer in solar cell, HPs made a dynamic entry into photovoltaic applications [5]. These materials have been extensively studied and employed for solar cells, LEDs, photodetectors, gas sensors, switching devices, FETs, metal-ion batteries etc. In 2016, the most promising Pb-HP perovskite made its debut into solar-driven hydrogen generation [6]. However, realizing the inevitable stability and toxicity issues associated with lead HP, there have been constant efforts to develop sustainable and stable HP perovskites for renewable hydrogen generation.

In the current review, we summarize Pb-free HPs for solar-driven hydrogen generation (evolution) derived from aqueous solutions. As a part of this discussion, a general introduction to solar-driven hydrogen generation systems, including status and issues associated with Pb-HPs in hydrogen generation, is elaborated. A detailed discussion is provided on the novel Pb-free HP compositions developed for hydrogen generation strategies to improve the performance well as the stability of these materials.

## 2. Solar-Driven Hydrogen Generation Systems

Figure 1 shows the HP-based water splitting process and systems. Conventionally solar-driven hydrogen generation systems are classified into three types of systems based on the operating mechanisms—photocatalytic, photoelectrochemical (PEC), and photovoltaic-powered electrolysis (PV-E) hydrogen generation.

Figure 1a summarizes the basic principle of photocatalytic water splitting. Splitting water into its constituent H_2_ and O_2_ molecules via a photocatalytic process requires a visible-light-responsive photocatalyst with a sufficient potential to overcome the positive Gibbs free energy from the splitting reaction of the water. Therefore, the photocatalyst should have a suitable, sufficiently narrow bandgap to harvest visible photons, and its bandgap and band edge position must encompass both the water reduction and oxidation potentials [7,8]. Conduction band minimum (CBM) of the photocatalysts should be more negative than the reduction potential of water (H_2_O to H_2_ at +0 V vs. NHE) to allow photogenerated electrons to effectively transfer to H_ad_^+^. At the same time, the valence band maximum (VBM) of the photocatalysts should be more positive than the oxidation potential of H_2_O to form O_2_ (+1.23 V vs. NHE). Generally, the photocatalysis mechanism involves three major steps: (1) photogeneration of charge carriers (electron–hole pairs) which can be enhanced by bandgap tunability and high absorption coefficient. (2) Diffusion of photogenerated charge carriers towards the redox-active sites on the surface. Charge separation and collection efficiency can be enhanced with high carrier lifetime, and defect and surface passivation hetero/homo-junctions, ensuring that more electrons and holes successfully participate in the desired redox reactions. These systems are characterized by a two-step excitation and/or charge separation process and mimic the natural photosynthesis process, harnessing a wide range of visible light for splitting because a change in Gibbs free energy (*∆G*_0_ = 237.18 kJ mol^−1^) required to drive each photocatalyst can be reduced, as compared to the one-step water splitting system and that the separation of evolved H_2_ and O_2_ is possible. (3) The final step is photocatalytic oxidation and reduction of water molecules for the evolution of hydrogen and oxygen. As the photocatalyst is dispersed in aqueous solutions, the stability of these materials plays a crucial role with limited choices.

Figure 1c shows the schematic of the PEC water splitting system, consisting of a photocathode and photoanode immersed in an aqueous solution, where the half-cell reactions constitute oxidation at the anode and reduction at the cathode, resulting in water splitting into H_2_ and O_2_. The performance of the PEC water splitting system is quantified by solar-to-hydrogen (STH) efficiency (*η_STH_*), usually measured under a standard solar irradiance of AM 1.5 G with no applied bias. *η_STH_* is calculated from the photocurrent density generated *J_sc_* (in mA cm^−2^) and Faradaic efficiency (*η_F_*) contribution to the hydrogen evolution reaction (HER), and 1.23 V represents the standard potential required for the electrolysis of water. Alternatively, H_2_ production/evolution can be measured by gas chromatography or mass spectrometry.
$$\eta_{STH}=\frac{\left(J_{sc}\times 1.23\times \eta_{F}\right)}{P_{in}}\times 100$$

Figure 1d shows the schematic of a PV-E-based water splitting system wherein a PV device provides the required potential to drive oxidation-reduction reactions for H_2_ evolution.

In this section we briefly classified conventional solar-driven hydrogen generation systems into three types based on the operating mechanisms: photocatalytic, photoelectrochemical (PEC), and photovoltaic-powered electrolysis (PV-E) hydrogen generation. In each type of system, there are more variations in the device configurations. Readers can refer to the recent review article, which summarizes more specified and idealized device configurations in water splitting, such as tandem and monolithic devices [9].

## 3. Classification of HP Perovskites Based on Their Structure

Based on the ionic radius of the cations and anions, perovskite materials adapt either of the following structural dimensionalities: 0-D or 2-D, or 1-D or 3-D structures. Figure 2 shows the structure of different dimensional perovskites. Considering a 3-D perovskite with a general formula of AMX_3_, a material that has continuous corner-sharing metal halide [MX_6_]^4−^ octahedra and a 3-D perovskite can be transformed into a 2-D perovskite by incorporating a spacer cation which separates the metal-halide octahedra. Whereas 1-D perovskites consist of either face-sharing or corner-sharing octahedra and, in the case of the 0-D perovskites, the metal-halide octahedral clusters exist individually and are surrounded by the A-type inorganic or organic cations. Considering the strictly periodical spatial arrangement, the metal-halide octahedra and wrapping/surrounding of the A-type cation species of the octahedra, the lower dimensional (2-D, 1-D, and 0-D) perovskites can therefore be treated as assemblies of 2D quantum wells, 1D quantum wires, or 0D molecules/cluster. Thus, lower dimensional perovskites are structurally unique and usually differ from morphological forms of 3-D perovskites like nanosheets/nanoplatelets, nanowires/nanorods, and nanoparticles/quantum dots based on 3D AMX_3_ [10]. For example, the 1-D perovskites morphologically resemble nanorods or wires with a strong interaction between the metal-halide species. These interactions lead to the formation of the electronic band formation yet, with the material being limited in length, they end up favoring the quantum confinement effect [11]. Contrary to this, in a molecular level 1-D perovskite, anionic metal halide species are surrounded by organic cations and are isolated from one another. This framework results in bulk assemblies of quantum wire-like structures, that is, macroscopic crystals that exhibit properties of nanomaterials. However, lead-based 1-D perovskites did not progress greatly and therewith Pb-free 1-D perovskites have not been investigated extensively [12,13]. For instance Chenkun et al. attempted to synthesize 1D (C_4_N_2_H_14_)SnBr_4_ alongside 0D (C_4_N_2_H_14_Br)_4_SnBr_6_ and observed photoinduced degradation into 0-D perovskite [14].

Furthermore, the dimensionality of the perovskite plays a crucial role in the stability, electrical, and optoelectronic properties of the material. Typically, 3-D perovskites with small (alkyl chain organic) cations are susceptible to moisture, temperature, and/or photo–induced degradation. Incorporating a bulky/long chain organic cation into a 3-D perovskite matrix not only transforms it into a lower dimensional 2-D perovskite, but also improves the air/environmental stability of the materials. Higher formation energy and hydrophobicity of the A site cations of the lower dimensional perovskites, particularly 2-D perovskites, offer greater stability over 3-D counterparts. On the other hand, 0-D perovskites, owing to their isolated and undisturbed metal-halide cores, offer the highest stability; in fact, unstable tin-based perovskites subjected to photodegradation or oxidation of Sn end up stabilizing into stable 0-D perovskites [14,15]. To understand the electronic and optoelectronic properties, it is essential to acknowledge the origin of the electronic band formation. Generally, with a reduction in the dimensionality of the bandgap and excitonic binding, energy increases, which hinders their application in solar cells. However, with the quantum confinement(-like) effect due to electronic bands generated by the alternative A-type cation spices and metal-halide species, lower dimensional (particularly 2-D) perovskites are prominent in LED and photodetector applications. For a perovskite, the conduction band and valency band are formed from the hybridized orbitals of the metal cation and halide anion, with little to no contribution from the A-type (organic and/or smaller) cation; with the reduction in the dimensionality, the connection between the metal-halide framework is weakened, which is the basic reason for the increase in the bandgap, as well as the reduction in the electronic dimensionality, otherwise low charge mobility (higher effect mass of the charge), of the material. Despite this, the electronic dimensionality and effective mass of the charges of the perovskite or even the bandgap can be improved by wisely choosing/replacing the inactive A-type cation otherwise orienting/tilting the metal-halide octahedra [16,17]. This viability 2-D as well as 0-D perovskites is greatly exploited in tuning these materials for specific applications [18,19]. 

Apart from the classification of perovskite in terms of the structural dimensionality perspective, replacement of the Pb in different metal cations leads to the following perovskite/perovskite-inspired materials. When Pb^+2^ is replaced with trivalent metal cations (like Bi^+3^ or Sb^+3^), leading to the formation of perovskite-inspired materials, which are often structurally 0-D and have a general formula of A_3_B_2×9_; these 0-D perovskites can be structurally tuned to form a layered 2-D perovskite as well; Lee et al. introduced polyethylene glycol (PEG) ligands to a 0-D Cs_3−n_Bi_2_X_9_ to synthesize a 2-D [(PEG_6_^−^NH_3_^+^)_n_Cs_3−n_Bi_2_X_9_, perovskite [20]. On the other hand, when Pb^+2^ is replaced with a pair of metal cations, M^+1^ and M^+3^ form the vacancy-ordered double perovskites with a general formula of A_2_M(I)M(III)X_6_. Additionally, other types of vacancy-ordered perovskites feature metal cations paired with tetravalent metal cations and follow the general formula A_2_MX_6_. Tin-based perovskites (Sn-HPs), specifically Cs_2_SnX6, are well-known example of this category, noted for their air stability and applications in solar cells and photocatalysis. 

## 4. Lead Halide Perovskites for Hydrogen Generation

In 2016, Park et al. [6] pioneered the use of HPs for hydrogen generation through hydroiodic (HI) acid splitting, by leveraging the dynamic equilibrium of ions in methylammonium lead iodide (MAPI) perovskite in aqueous HI and HBr reaction mediums, as shown in Figure 3a. The study demonstrated continuous photocatalytic activity of MAPI for approximately 6.5 days, showcasing robust phase stability and potential for hydrogen generation. Dissolution of the PbI_3_^−^ ions in aqueous solution requires controlled accessibility of I^−^ ions, while an excess of I^−^ ions in the reaction medium leads to the formation of PbI_3_^−^ and PbI_4_^−2^ ions. The availability of H^+^ ions is to be monitored to restrict the formation of hydrated phases and induce forced precipitation of MAPI crystals. Therefore, MAPI, in aqueous solution, forms hydrated phases or disassociates into its constituent metallic salt and requires a specific range of concentration of H^+^ and I^−^ ions for precipitation and stabilization. This understanding led Park et al. to employ aqueous HI and HBr reaction mediums to produce hydrogen gas by photocatalytic activity of MAPI. The study highlights a continuous photocatalytic activity of MAPI under illumination for about 6 and a half days. MAPI exhibited robust phase stability due to the dynamic equilibrium of the material and the saturated reaction system. A series of experiments, conducted under variable conditions, including variable intensities and wavelengths of visible light, and in the presence of the Pt co-catalyst and an H_3_PO_2_ reducing agent, indicated the potential of employing MAPI in an aqueous environment for hydrogen generation by its photocatalytic activity.

Subsequently, a series of mixed halide perovskites were utilized for hydrogen generation from aqueous hydrohalic acids. Wu et al. significantly enhanced photocatalytic hydrogen evolution activity by incorporating a Br ion into MAPI [21]. They prepared powdered samples of MAPbBr_3−*x*_I*_x_* with a gradient of iodide concentration, gradually increasing from the interior to the surface of each particle to a facile light-assisted halide exchange, resulting in a bandgap funnel structure, as shown in Figure 3b(i). The distribution of Cl and I ions within the perovskite crystal is as shown in Figure 3b(ii). In this unique architecture, photoluminescence (PL) experiments revealed swift charge carrier transport from the bromide-rich region to the iodide-rich region. The iodide-rich sites acted as the primary locations for charge recombination, while the bromide-rich sites served as the absorbing and carrier generation sites. This funnel structure effectively minimized carrier recombination, leading to improved H_2_ photogeneration. To demonstrate this, the experiments were conducted under aqueous mixed HI/HBr acid-saturated solutions, along with H_3_PO_2_ selectively reducing I^−^ ions that could interfere with light absorption. Under visible light irradiation (*λ* ≥ 420 nm, 100 mW cm^−2^), MAPbBr_3−*x*_I*_x_* exhibited H_2_ evolution of 1021.20 μmol g^−1^ h^−1^ (compared to 11.20 μmol g^−1^ h^−1^for pristine MAPbBr_3_). This efficiency was further enhanced to 2604.80 μmol g^−1^ h^−1^ by incorporating a 3 wt % Pt loading to separate electron–hole pairs. The solar-to-light chemical efficiency at the start of the reaction without H_3_PO_2_ was found to be 1.05%, and the apparent quantum efficiency of the MAPbBr_3−*x*_I*_x_*/Pt system under 450 nm irradiation reached 8.10%. Remarkably, this system demonstrated excellent stability over 30 h of continuous experiments, repeated in six cycles of 5 h each [21]. The authors extended their bandgap funnel approach to explore all-inorganic mixed perovskite, specifically the CsPbBr_3−*x*_I*_x_* system, using the same preparation method as before [22]. The bandgap alignment and photographic images of the synthesized perovskites are shown in Figure 3c(i,ii). In this case, CsPbBr_3−*x*_I*_x_* was also loaded with Pt co-catalyst. Under visible light irradiation (λ ≥ 420 nm, 120 mW cm^−2^), in an aqueous HBr solution saturated with CsPbBr_3_, the CsPbBr_3−*x*_I*_x_*, the system demonstrated remarkable H_2_ photogeneration, reaching 896.0 μmol g^−1^ h^−1^, while maintaining high stability for up to 50 h. The apparent quantum efficiency for this system was found to be 2.15% under 450 nm irradiation. [22]

Employing composites, a traditional strategy to improve photocatalytic activity of semiconductors, is adapted by research in the case of halide perovskites as well. Wu and his team developed a composite of MAPI with reduced graphene oxide (rGO) by dispersing GO powders in an aqueous HI acid solution saturated with MAPbI_3_. [23] Upon exposure to visible light (*λ* ≥ 420 nm) during the photoreaction, GO was reduced to rGO, as confirmed by Raman spectroscopy. Further analyses using Transmission Electron Microscopy (TEM) and Fourier transform IR (FTIR) validated the attachment of rGO to MAPbI_3_ and the formation of new chemical bonds between them. Figure 3d shows the SEM image of the MAPI/rGO; the inset shows the schematic of photocatalytic activity involved. Photocatalytic H_2_ evolution reactions were carried out on powder samples of the MAPbI_3_/rGO composite in an aqueous HI solution saturated with MAPbI_3_ (along with H_3_PO_2_, in order to reduce I_3_^−^). Compared to pristine MAPbI_3_, the composite exhibited a remarkable 67-fold improvement in photoactivity, reaching a rate of 939.90 μmol g^−1^ h^−1^. Interestingly, the authors found that rGO performed even better than Pt as a cocatalyst, as measured by hydrogen photogeneration in Pt-loaded MAPI. Moreover, the composite displayed excellent stability over 20 cycles of repeated experiments, with each cycle lasting 10 h. Photoluminescence (PL) measurements confirmed that the composite facilitated more rapid charge transfer and effective electron−hole separation, leading to an increased H_2_ evolution rate. [23]

Wang and colleagues employed a similar strategy to enhance HI splitting by MAPbI_3_, using Pt/TiO_2_ nanoparticles as nanoscale electron-transporting channels to efficiently extract electrons when in contact with MAPI [24]. To improve the interaction between the metal halide perovskite (MHP) and Pt/TiO_2_ nanoparticles, they heated the solution containing both semiconductors to promote MAPI dissolution and subsequent reprecipitation during cooling. This process strengthened the dynamic attachment of MAPI to Pt/TiO_2_, facilitating their interaction. Remarkable enhancement in hydrogen evolution was achieved with a 50% loading of Pt (0.75 wt %)/TiO_2_, resulting in an 89-fold improvement compared to pristine Pt/MAPI and leading to the evolution of 436.6 μmol of hydrogen in 6 h. The composite’s stability was confirmed over four successive photocatalytic cycles. Based on photoluminescence (PL) measurements, the authors proposed a mechanism in which MAPI particles dispersed in a saturated solution acted as light absorbers, generating electrons and holes upon light irradiation. The proper band alignment between Pt/TiO_2_ and MAPbI_3_ facilitated the transfer of electrons generated on MAPI to Pt/TiO_2_, where they reduced protons to form H_2_. Simultaneously, the holes on MAPI oxidized I^−^ to I_3_^−^, enhancing HI splitting efficiency. The authors also experimented with a composite using commercial Ta_2_O_5_ and Nb_2_O_5_ nanoparticles loaded with Pt. While Ta_2_O_5_ showed some effectiveness in promoting H_2_ evolution, its potentially unfavorable band alignment resulted in negligible catalytic activity compared to TiO_2_. However, it is important to note that these findings were preliminary, and further investigations could be pursued to explore the potential of such composites [24]. 

Recently, Jiang et al. optimized the surface of three-dimensional perovskite (MAPbI_3_) films by incorporating a two-dimensional perovskite layer [25]. They utilized FTO/glass as a waterproof hole-transport layer to encapsulate the cell. These resulted in a reduction of defects and optimized energy-band alignment, leading to a halide-perovskite photoanode with a high-efficiency oxygen evolution reaction and exceptional long-term stability. The photoanode maintained 90% of its initial photocurrent density (16.55 mA cm^−2^ at 1.23 V vs. RHE) for 30.9 h under solar illumination [25]. Fu et al. developed a system utilizing PMA_2_PbI_4_/MoS_2_ as the photocatalyst for hydrogen evolution and WO_3_/RuO*_x_* as the photocatalyst for oxygen evolution [26]. This system achieved hydrogen and oxygen production in a stoichiometric ratio with a solar-to-hydrogen (STE) efficiency of 2.07%. Hansora et al. applied a nickel foil and nickel–iron oxyhydroxide (NiFeOOH) on α-phase FAPbI_3_ as passivation layer and oxygen-evolving catalyst (OEC), respectively, to fabricate the NiFeOOH/Ni/FAPbI_3_ photoanode for water oxidation [27]. They achieved relatively high STE efficiency, ranging from 9.8% for a small-area device (0.25 cm^2^) to 8.5% for scaled-up photoanodes (123.2 cm^2^) under 1-sun conditions. Additionally, the device maintained good stability for 2 h. Jeong et al. also focused on scaling up photoelectrode devices for solar hydrogen production. They used cyclohexylammonium iodide (CHAI) to regulate nucleation kinetics and passivate grain boundaries. This approach enabled the fabrication of 8 cm × 8 cm coplanar LHP-based photoelectrodes that could operate under parallel illumination without any applied bias. These photoelectrodes achieved a record solar-to-hydrogen (STH) efficiency of 9.89% and maintained 80% of their initial photocurrent density for 24 h [28]. 

While traditional PV devices suffer from low open-circuit voltages and do not overcome the thermodynamic driving forces and overpotentials of the water splitting reactions, HP solar cells possess sufficiently high V_oc_s advantages for water splitting [6,29,30,31]. However, integration of PV systems with the electrochemical cell is a challenging task. Designing multijunction/tandem junctions is an effective strategy to improve power conversion efficiency of the solar cells. Combining wide bandgap HP with a narrow-bandgap crystalline Si for tandem solar cells recently surpassed the Shockley Queisser limit, and researchers from KAUST and Helmholtz Zentrum Berlin’s (HZB) demonstrated perovskite-Si tandem solar cells with efficiencies beyond 32%. With the PV-assisted water splitting, which requires high operating potential, these devices with high open-circuit voltage (V_oc_ > 1.23 V) are of special interest. For the first time Dutta et al. integrated an electrochemical cell and a monolithic lead HP-Si tandem solar cell to achieve solar-to-hydrogen (STH) efficiency > 21% [30]. With electrochemical cells operating at high photocurrent of 18 mA and an operating potential over 1.4 V, which is close to the maximum power point of the employed solar cell, the designed water splitting system is conveniently without light-concentrating techniques [30]. K. Fehr et al. developed a durable HP photochemical cell fabricated with a conductive adhesive-barrier (CAB) [31]. The CAB protected the HP from corrosion and degradation during the water splitting reaction. Stability and solar-to-hydrogen (STH) efficiencies were demonstrated for two electrochemical systems: one featuring a co-planar photoelectrode and the other utilizing an HP-Si photoanode. For a co-planar photoelectrodes system, the researchers achieved a solar-to-hydrogen (STH) efficiency of 13.4% and a lifetime of 16.3 h to reach *t*_60_ (60% of initial photocurrent). A monolithic silicon–perovskite tandem photoanode outperformed its counterpart with a remarkable STH efficiency of 20.8% and a lifetime of 102 h to reach *t*_60_ [31]. Yang et al. fabricated efficient and stable PV–PEC tandem devices comprising low-cost Sb_2_Se_3_ photocathodes and semitransparent perovskite. The optimum tandem device achieved a photocurrent density of 8.3 mA cm^−2^ without external bias and STH conversion efficiency exceeding 10%. The device maintained 80% initial photocurrent value after a 10 h test [32]. The same group also reported another wireless artificial leaf design with a FAMACsPbI_3_-based dual PV module as the photocathode and NiFe-OH as the catalyst anode. The photocathode exhibits an excellent stability of 85 h, and the artificial leaf device exhibited an average STE efficiency of 11.4% [33]. 

## 5. Issues with Lead HPs for Water Splitting

### 5.1. Toxicity

Halide perovskite energy conversion systems, including solar cells and water splitting systems, demonstrate superior power conversion efficiency compared to traditional systems. However, it is worth noting that the most efficient PSCs are constructed using lead halide salts (PbI_2_ and PbBr_2_), which renders them potentially hazardous due to their toxic nature. Figure 4 shows the possible ways of lead leaching into ground water, followed by entering the food chain and eventually into human intake [34]. Based on energy market projections for 2050, we can estimate the required amount of lead to produce these solar cells. Ram et al. conducted calculations and predicted that the electricity energy demand from PVs in 2050 would be 12,210 TWh/year, accounting for 60% of the total electricity demand [35]. Assuming that PSCs will contribute approximately 30% of the PV energy share, the demand for lead for PSCs would amount to 1000 tons per year for the European Union (EU) alone [35]. 

Despite the amount of lead used in PSCs being comparably low (as compared to other applications), there is a concern arising from the relatively high water solubility of lead halide salts, which is higher than other heavy metal compounds commonly employed in the PV industry, such as CdS, PbS, and CdTe. For instance, the water solubility of PbI_2_ is approximately one billion times higher than that of PbS and PbSe [36]. This solubility poses a problem, as water/moisture (as well as oxygen) can cause damage to solar cells, leading to the decomposition of perovskite into PbI_2_, hydroiodic acid, and methylamine [37]. The interaction between MA^+^ and H_2_O forms a strong hydrogen bond, weakening the original bond between MA^+^ and the Pb-I octahedral. This accelerates the deprotonation of the organic cation, and the proton can transfer to I^−^ via H_2_O, resulting in the production of volatile species like CH_3_NH_2_ and HI leaving behind PbI_2_ on the device [38]. Particularly, failure of encapsulation of the photoanode leads to the leaching out of lead into ground water ingress and is associated with elevated bioavailability, which means a mere 10% increase in lead contamination from perovskite in soil could result in a staggering over 100% increase in lead contamination within plants. Consequently, plants grown in soils contaminated with lead, even at concentrations within stringent international limits for agricultural soils, exhibit evident lead intoxication [34,36]. 

### 5.2. Stability 

Intrinsically, the formation energy and tolerance factors, including Goldschmidt’s tolerance factor (*t*), Bartel’s tolerance factor (*τ*), and the factor of octahedra (*μ*), govern the structure and stability of perovskite. Empirical observations indicate the optimal parameters for 3D perovskite (ABX_3_) lie within the range of 0.8 < *t* < 0.9, 0.442 < *μ* < 0.895, and *τ* < 4.18 [7]. Extrinsically, factors like moisture, oxygen, and light also have great influence on the stability of perovskite. Photo-energy conversion systems used in practical applications must be designed to withstand diverse weather conditions, with temperature and humidity being critical factors. The International Electrotechnical Commission (IEC) standard includes three stability tests: thermal cycling, damp heat, and humidity freeze, which closely relate to these two factors. Ensuring stability against temperature and humidity can be assessed by considering various aspects, including the perovskite material, functional layers, interfaces, and back electrodes. These elements collectively contribute to the overall performance and durability of solar cells under challenging environmental conditions. However, popular lead halide perovskites, including MA-based and FA-based ones, face severe thermal instability [39]. Thermogravimetric analysis reveals that organic lead halide perovskites exhibit a high decomposition temperature beyond 85 °C, questioning their application for terrestrial photovoltaics. Reports indicate that decomposition can occur at 85 °C due to the volatilization and loss of various organic species in an open space [40]. Using less volatile alternatives, like FA^+^ and inorganic Cs^+^, instead of MA^+^ is a common approach. However, the non-photoactive yellow *δ*-phase of FAPbI_3_ and CsPbI_3_, rather than the black phase, is thermodynamically stable at room temperature [41]. 

#### 5.2.1. Moisture-Induced Degradation

Despite their popularity, phase transitions, instability to thermal stress, and exposure to ambient air and moisture have been constantly viewed as potential barriers for the organic–inorganic lead halide perovskite to be adopted for technology. Earlier in 2014, researchers identified the degradation of CH_3_NH_3_PbI_2_ into aqueous HI and methylamine and, followed by the interference of O_2_, the HI decomposes into I_2_ and H_2_O [42,43]. Figure 5a shows the proposed decomposition mechanism of MAPI in the presence of water molecules. Further, Snaith et al. established the roots of degradation of the perovskite to the dangling hydrogen bond and escaping of the MAI from the lattice structure, leaving behind PbI_2_ [44]. While Yang et al. investigated the degradation of MAPI using in situ GIXRD and absorption and found out the first step to degradation is the formation of perovskite dihydrate containing isolated PbI_6_^4−^ octahedra [45], Frost et al. suggested a reversible degradation mechanism that involves the phase transition of the perovskite molecule into its monohydrate and successive breaking down into organic salt and PbI_2_ [42]. 

#### 5.2.2. Oxygen and Photo-Induced Degradation

In 2015, Aristidou et al. accounted for the degradation of CH_3_NH_3_I_3_ in moisture-free conditions to the oxygen present in the atmosphere [49]. Despite careful encapsulation, the material tends to decompose in the presence of oxygen and light; the photo-exited perovskite results in the generation of superoxide by transferring an electron to an oxygen molecule, and in return this superoxide attacks perovskite and decomposes into the organic and inorganic salts with water as a by-product. The decomposition can be summarized as follows:

Degradation mechanism of CH_3_NH_3_PbI_3_ in the presence of oxygen
$${\mathrm{MAPbI}}_{3}\overset{\begin{array}{c} \mathrm{Light}\;\\ \mathrm{No}\;\mathrm{Moisture} \end{array}}{\to }{{\mathrm{MAPbI}}_{3}}^{*}\;O_{2}\overset{{{\mathrm{MAPbI}}_{3}}^{*}}{\to }O_{2}^{.-}\;{\mathrm{MAPbI}}_{3}+O_{2}^{.-}\overset{\mathrm{Deprotonation}}{\to }{\mathrm{MeNH}}_{2}+{\mathrm{PbI}}_{2}+\frac{1}{2}I_{2}+H_{2}O$$
* photo-excited spices of MAPI perovskite. 

The following year, the same group identified that, contrary to reports suggesting adverse effects of moisture-induced degradation, the rate of light and oxygen induced degradation is higher. Figure 5b shows the degradation of MAPI under various conditions and, clearly, the degradation of the material in dry air and open air under light condition does not exhibit much difference, indicating that moisture is not the only major culprit [46].

Tang et al. studied the photodegradation of the perovskite thin film under various conditions (in nitrogen, vacuum, and air) and unveiled the photoinduced degradation of the CH_3_NH_3_PbI_3_ under various experimental conditions (N_2_, vacuum, and atmospheric conditions under light and dark environments) [47]. Figure 5c shows the XRD patterns of MAPI films exposed to various conditions. The light-exposed sample under vacuum shows no sign of absorption, ascertaining the detrimental effect of photons on the perovskite. This investigation identified the similarity of the photodegradation of CH_3_NH_3_PbI_3_ and photolysis of PbI_2_. Previously discussed reports suggest that the degradation of MAPI under vacuum conditions primarily involves the formation of non-ionic molecular (MAI) defects, rather than ionic defects, and imply that the loss of the CH_3_NH_3_^+^ cation is closely linked to the sequestration of iodide from the anion octahedron lattice (PbI_6_)^4−^, thereby creating iodine vacancies within the MAPI films. This study highlights that the initiation of photoinduced degradation in MAPI within a vacuum arises from the generation of iodine vacancies due to MAI loss [47].

Sun et al. studied the photo-induced degradation of perovskite processed using two different precursors, Pb.Ac_2_ (Ac being acetate) and solution-engineered (sol-eng) processed perovskite films, and co-related the diffusion of O_2_ and variation in the microstructure of the material [48]. Interestingly sol-eng perovskite showed faster degradation compared to the Pb.Ac_2_ perovskite and this was attributed to the small gains of the sol-eng perovskite with high defect density at the boundaries, which thereby makes the material more susceptible to the oxygen diffusion (Figure 5d). On the other hand Pb.Ac_2_ perovskite is characterized by large gains and O_2_ attacks on the grain boundaries, followed the propagation of the defects throughout the material. Figure 5d shows the SEM images of the Pb.Ac_2_ perovskite and diffusion for O_2_ across the grains and the relative degradation mechanism [48].

### 5.3. Bandgap of Pb-HPs and Need of Pb-Free HPs in PECs

Conventional lead perovskite with iodide halide has a narrow bandgap of 1.54 eV and is known to deliver high saturation current. However, the material is not an ideal choice for water splitting due to the photoelectric voltage which barely supports the water oxidation and charge injection potentials. On the other hand, chlorine-based perovskites are limited by the low absorption spectrum. Alternatively, bromide perovskites exhibit suitable energy bandgaps and are apt for the designing of low-onset potential photoanodes. Particularly, cesium lead bromide (CsPbBr_3_) has the larger bandgap at 2.4 eV, yet the lowest theoretical saturation current, which required combating strategies to compensate for the saturation current. Moreover, while all inorganic perovskites require high temperature synthesis, methylammonium lead bromide (MAPbBr_3_) has poor thermal and moisture stability. This opened doors for formamidinium lead bromide (FAPbBr_3_) as a candidate material for photoanodes for water splitting. Recently, Yang et al. investigated a monolithic FAPbBr_3_ photoanode for water splitting. The photoanode delivered high photon-to-current efficiency of 8.5% with an open-circuit voltage of 1.4 V. Interestingly, the energy band offset allowed the coupling of the photoanode with a carbon/graphite protective layer, improving the stability of the system to an operational 100 h [50]. On the other hand, 2D perovskites inherently have higher bandgaps, which ensure sufficiently high potential to facilitate water oxidation/splitting. For example (HDA)_2_PbI_4_ perovskite, a lead compound with hexadecylammonium (HDA) organic cations incorporated between the metal halide nanosheets, exhibited a bandgap of 2.4 eV and offered high stability in a water environment because of the hydrophobicity of the long chain organic molecule [51]. Similarly, long chain alkyl amine perovskites (C_n_H_2n+1_NH_3_)_2_PbI_4_ (*n* = 14, 16, and 18) have an absorption edge extended to ~540 nm and demonstrated excellent stability up to 96 h when dispersed in water [52].However, lead-based 2D perovskites for water splitting have not been extensively explored, probably due to the limited carrier mobility governed by low structural dimensionality.

While instability of conventional (organic) lead halide perovskite is a significant problem that needs to be addressed, some progress has been made in moving towards lower-dimensional perovskites as well as all-inorganic perovskites. Proposing the development of 2D/3D heterojunction perovskites that could provide a trade-off between bandgap, stability, and charge mobility emerges as a potential solution [53]. However, the toxicity concern primarily centers around lead, and efforts have been made to identify alternative materials that can substitute it. 

## 6. Lead-Free Perovskites in Water Splitting/Hydrogen Generation

### 6.1. Structure and Bandgap in (Lead-Free) Perovskite

Consider a 3-D perovskite AMX_3_ such as CsSnX_3_ (X = I, Br, Cl); Huang et al. studied the electronic band structure of the material using the quasiparticle self-consistent GW method (QSGW) [54]. For a cubic phased perovskite, the VB maxima is comprised of the mostly hybridized X’s ns states and a band which is antibonding between Sn 5s and X’s np orbitals and the CB minima is mainly comprised of the Sn’s 5p states. Furthermore, conduction band minima (CBM) exhibit threefold degeneration without SO coupling and split into a doublet and a quadruplet (including spin) when SO is included [54]. Similarly for a 3-D MASnX_3_ and 2-D perovskite, the contribution from the larger (organic) cation is insignificant. In fact, the hybridized orbitals of the Sn-s to X-p form strong band dispersion at the valence band maxima (VBM) and lead to higher hole mobility, thus the material was first reported as a solid hole transport layer for the dye-sensitized solar cells [55]. However, without spin-orbital coupling and band splitting at the CBM, the electron mobility falls short compared to conventional Pb-perovskites. Moreover, the stabilization of tetravalent ions of the metal ions increases from Pb to Sn to Ge to Si over their divalent forms, which argues for the synthesis and applications of the 3-D Ge and Si perovskites. 

The conventional method to tune the bandgap by partial to full replacement of halide anion and the A-type cation in the traditional lead-based perovskite is equally effective for the 3-D lead-free perovskites. Typically, for a 3-D tin-based perovskite, MASnCl_3_, the bandgap is reported to be 1.1 eV and replacement of MA with FA or Cs led to increasing the bandgap to 1.4 and 1.36 eV, respectively [56]. Although the literature suggests the non-involvement of the A-type cation in the CB and VB formation, the A-type cation is responsible for the titling of the metal-halide octahedra, thereby affecting the bandgap [16]. Other arguments validated the signature of the C- and N-orbitals of the organic cation in the formation of the VB [57]. Replacement of MA with a larger cation, like dimethylammonium, not only increased the bandgap (2.05 eV to 2.9) but also transformed the crystal structure from pure cubic to orthorhombic [58].

The exploration of group-VA cations (M^3+^ = Bi^3+^ and Sb^3+^) as potential replacements for Pb^2+^ aims to maintain the chemistry of the lone-pair ns^2^ state, which is known for its benefits in achieving high photovoltaic performance. However, due to their higher +3 oxidation state, the normal AMX_3_ perovskite structure, consisting of corner-sharing [MX_6_] octahedra, cannot form. Instead, A_3_M_2_X_9_ emerges as the stable stoichiometry in this context. Generally, researchers have experimentally reported two main phases of A_3_M_2_X_9_: the hexagonal phase, featuring zero-dimensional (0D) bi-octahedral face-sharing clusters [M_2_I_9_]^3−^ (also known as the dimer phase), and the phase composed of two-dimensional (2D) corrugated layers with partially corner-sharing MX6 octahedra (referred to as the layered phase). The 0D dimer phase can be easily synthesized through low-temperature synthesis methods.

Perovskites with two metal cations have the general formula A_2_M(I)M(III)X_6_ or A_2_M(IV)X_6_, the latter one also known as vacancy-ordered perovskites. The A cation is typically a large monovalent ion, such as Cs^+^. The M(I) and M(III) cations are often transition metal ions, like Ag^+^ (or K^+^, Tl^+^) combined with Bi^3+^ (or Sb^3+^). Alternatively, the cation can be a simple tetravalent metal ion like Sn^4+^ (or Te^4+)^ in vacancy-ordered perovskite. The crystal structure of double perovskites can be visualized as a framework of corner-sharing octahedra. The larger A-site cations occupy the voids between the octahedra. The smaller metal cations are located at the center of the octahedra. The octahedra share their corners, forming a three-dimensional network. This arrangement provides structural stability for the material. The double perovskite structure is highly flexible, allowing for the incorporation of different combinations of cations at the M(I) and M(III) sites. This flexibility gives rise to a wide range of properties and functionalities in double perovskite materials. Furthermore, in A_2_M(IV)X_6_, the octahedron is no longer connected by the halide anion resulting from the alternative arrangement of vacancies and M(IV) cations. Generally, considering their valence atomic orbitals, the possible substitute for metal cations can be metals (1). with the semi-core d states, (2). with the lone pair s state, and (3). without the d or s states. And the presence of a lone pair due to the ns^2^ electrons from the metal cation determines the nature of the bandgap. For instance, the predominant Ag-based double perovskite systems Cs_2_AgBiX_6_ exhibit indirect bandgaps with a mismatch of the CBM and VBM at the L point and X point, respectively, and angular momentum mismatch of the orbitals that comprise the CBM and VBM. Moreover, with M(I) being monovalent and M(III) being a trivalent metal cation, the band edges are determined by the [M(III)X_6_] octahedra rather than M(I). The CB and VB arise from the np orbital of the M(III) and antibonding orbitals of the M(III)’s ns and X’s np orbitals. This leads to the isolation of the [B(I)X_6_] octahedra and is responsible for the electronically 0-D structure. Thus, the indirect and large bandgap, alongside the reduced electronic dimensionality leading to the relatively large carrier effective masses and reduced mobility, makes the Cs_2_AgBiX_6_ a less-than-ideal choice for solar cells. However, the material has significant importance in LEDs and in recent times has been explored for photocatalysis.

In case of double perovskite A_2_M(I)M(III)X_6_, VBM is formed by bonding orbitals of MIII (ns), MI (nd), and X (np), whereas the conduction band minimum (CBM) is formed by antibonding orbitals MIII (np) and X (np). This electronic band structure is similar to that of a 3-D perovskite as well as GaAs or CdSe [59]. On the other hand, absence of antibonding orbitals at either of the band edges leads to poorer defect-tolerant materials. To understand the electronic band of a 3-D double perovskite, let us consider a novel Bi–Ag perovskite system, say Cs_2_AgBiX_6_. The outer shell electronic configuration of Bi^3+^ is 6s^2^6p^0^ and is the same as Pb^2+^. VB is predominated by the X[np] and the Ag[4d], while the CB is formed by X[np], Ag[5s], and Bi[6p] states. Further, the mismatch in the localization of the CBM and VBM gives rise to the indirect bandgap in these materials. An ordered Cs_2_AgBiBr_6_ exhibits an indirect bandgap of 2.04 eV and can easily be switched to direct bandgap disordered Cs_2_AgBiBr_6_. It is known the direct bandgap materials have a higher mobility (lesser effective mass of electrons) compared to that of the counterpart with the same bandgap. Thus, the disordered Cs_2_AgBiBr_6_ has a suitable direct bandgap of 1.59 eV with higher mobility and is apt for photovoltaic and photocatalytic applications [60]. 

The stability of the A_2_M(I)M(III)X_6_ is mostly dependent on the cations A and M(I) and the halide X. The theoretical calculations suggest a perovskite system with larger A^+^ cations (e.g., Cs^+^ is preferred over Li^+^) and smaller halides (e.g., F^−^ is preferred over I^−^) has higher stability. Most investigated double perovskites contain Cs cation and either Cl or Br anions and, to the best of our knowledge, the bandgap of these systems is suitable for PV and PEC applications. For example, Cs_2_AgFeCl_6_ exhibited a bandgap of ∼1.55 eV and replacing the Fe with Ga lead to a reduction in the bandgap to ∼1.37 eV [61]. Also, stability is influenced by the size of the metal cation M(I) and often perovskite systems with Ag+ have higher stability compared to Cu^+^. Introducing larger cations, like iodine, into the perovskite system for tuning the bandgap could be an unfavorable approach and can lead to a collapsing of the perovskite structure due to crystal instability. Interestingly, alloying of the trivalent metal cations can be adopted or finely tuned by the bandgap and switching between direct and indirect bandgap material. 

Furthermore, in the case of double perovskites, a rational substitution of the metal cations allows tuning from indirect to direct bandgap materials. The key is to choose M(I)/M(III) cations in A_2_M(I) M(III)X_6_ with a lone-pair in the ns orbital, resulting in strong s–p coupling between the ns orbital and X p orbitals, or induce a disordered metal alloy at the respective metal ion sites. Alternatively, double perovskites with metal cations fully occupied pseudo-closed s^2^ shells have a similar band structure as lead perovskite and pose a direct bandgap, high mobility, and high absorption coefficient. However, unlike Bi/Sb- Ag-based double perovskites, not many of these perovskites are realized. Zhao et al. reported design of the double perovskites for photovoltaic application, which reveals that Cs_2_InBiCl_6_ and Cs_2_InSbCl_6_ exhibits a direct bandgap, with the CBM formed from antibonding states of metal cations’ np orbitals and Cl 3p orbitals, and VBM from the antibonding states of metal cations’ ns orbitals and Cl 3p orbitals [62]. Theoretical studies identified that (MA)_2_TlBiBr_6_ shows a similar electronic structure as conventional lead perovskite, and experimental results confirmed the direct band of the material to be 2.16 eV. However, the presence of Tl, which is again a toxic element, is the major setback for this material [63].

Perovskites crystallizing into 2-dimensional units separated by interlayer species are termed layered perovskites. Depending on the spacer cations, the layered perovskites can fall in one of the two categories—Ruddlesden–Popper (RP) perovskite (with a chemical composition A_n−1_M_n_X_3n+1_) and Dion–Jacobson (DJ) perovskite (with a chemical composition F′A_n−1_M_n_X_3n+1_ where F and F′ are monovalent (+1) and divalent (+2) organic spacer cations, respectively). To simplify the understanding, RP perovskites have spacer molecules (generally significantly larger organic cations) interlayered between the edge-sharing inorganic octahedra to create a 3-D network, whereas DJ perovskites still have organic spacer layers but are connected in a zig-zag fashion with corner-sharing inorganic octahedra resulting in a 2-D framework. Interestingly, the structural stability of the RP perovskite is vulnerable due to variable thickness. On the other hand, DJ perovskites are limited by tolerance factors due to the restricted movement of corner-sharing species. Apart from these structural differences, both of the layered perovskites share similar traits. The metal-halide framework is mainly responsible for the optoelectronic properties; the spacer cations play a role in physical properties like exciton binding energy and charge mobility and exhibit a wider bandgap, better stability, and moisture resistance compared to their 3-D counterparts. It is worth noting that, in 1994, D.B. Mitzi, in his communications about synthesis of the Sn layered perovskite, commented about the tuning of electronic and magnetic properties in layered perovskites without disturbing the core active layers [64]. Thanks to the interlayer arrangement of the organic and inorganic layers in the layered/2-D perovskites, the material exhibits better overall stability because of the following factors: (1). restriction of the movement of the ions; (2). structural confinement and maintained integrity of the material; (3). restriction of the moisture ingress by the organic spacer layers with tunable length; and (4). tolerance to mechanical strain and thermal cycles. In fact, incorporating a small amount (0.08 M) of layered (Sn) perovskite in the 3-D FASnI_3_ perovskite improved the crystallinity of the material and reduced the trap states, which are otherwise caused by the oxidation of the Sn^+2^ to Sn^+4^ in 3-D tin perovskites. This 2D/3D hybrid tin perovskite devices exhibited 9% PCE (vs. 6% PCE for the 3-D Sn perovskite device) with enhanced light and environmental stability [65]. 

For 2-D Ruddlesden–Popper) layered perovskites, the inorganic metal-halide [MX_6_]^4−^ layers are separated by insulating organic/separator layers, and these separator layers are connected via van der Waals forces. Provided the thickness of the inorganic sheet is comparable to the de Broglie wavelength of the carriers, 2-D materials experience strong quantum confinement and quantum size effects [66]. This results in unique quantum well-like electronic band structures with a formation of strongly bound exciton and increased photon absorption [67,68]. Thus, 2D perovskites exhibit different optical properties compared to 3D ones. Evidently, the size and shape of the organic and the spacer cations determine the tilting and relative positioning of the [MX_6_]^4−^, interfering with the optoelectronic properties of the material. Madal et al. studied the influence of the positioning of the fluorine-substituted PEA in the phenyl ring of (PEA)*_2_*SnI_4_. However, the study revealed that the bandgap variation is insignificant [69].

By our understanding of 3-D, 2-D, and 0-D perovskite structures and structure–bandgap correlation and stability, researchers identified various potential halide-perovskite compositions for hydrogen generation. Based on the conduction and valence band potential levels with respect to the NHE, these lead-free perovskites are employed either in photocatalytic or photoelectrochemical systems. The following discussion provides a summary of Pb-free compositions used for hydrogen generation.

### 6.2. Photocatalytic Water Splitting Using Pb-Free Halide Perovskites

Figure 6 shows the bandgap alignment of various Pb-free HPs vs the NHE, highlighting those that showed promising results in solar-driven hydrogen generation from water or aqueous solutions.

#### 6.2.1. Bismuth and Antimony HPs

The stable oxidation state of bismuth and antimony (Bi^+3^ and Sb^+3^) is isoelectronic with Pb^2+^ with a comparable ionic radius and similar valence band configuration (6s^2^), making bismuth a potential replacement of Pb^2+^ in conventional perovskite. However, owing to the heterovalent metal replacement, the perovskite adapts into a 0-D structure with A_3_Bi(/Sb)_2_X_9_ formula. Thus, 0-D perovskites are often difficult to blend into a continuous uniform film and possess an indirect bandgap. These factors restrict their application in photovoltaic applications. However, they are promising lead-free halide perovskites for photocatalytic activities. Table 1 lists out the 0-D Bi and Sb HPs employed for photocatalytic water splitting.

Guo et al. (2019) successfully synthesized bright red colored bismuth perovskite MA_3_Bi_2_I_9_ using a simple solvothermal method for the photocatalytic hydrogen generation from an aqueous HI solution (hypophosphorous acid as a selective reducing agent) [70]. The perovskite powder has a VBM of −5.76 eV and a CBM of −3.78 eV, which are suitable for photocatalytic HI splitting. The researchers demonstrated the exceptional phase stability of MA_3_Bi_2_I_9_ in hydriodic acid (HI) aqueous solutions with varying concentrations under visible light irradiation. The photocatalytic activity of the material with the addition of Pt as a cocatalyst significantly enhanced the photocatalytic rate for H_2_ evolution, reaching approximately 169.21 µmol g^−1^ h^−1^, a 14-fold improvement compared to the bare perovskite. The solar chemical conversion efficiency was measured at 0.48%. The synthesized perovskite exhibited excellent phase and photocatalytic stability, even after 70 h of repeated H_2_ evolution the material remained robust in the reaction solution. The enhanced stability of MA_3_Bi_2_I_9_ was attributed to the oxidation state of bismuth (III) [70]. 

As photocatalytic activity takes place on the surface of the photocatalyst, nanostructure and morphology, which influence the surface area of the catalyst, and anisotropic properties, which vary based on the orientation of the catalyst, play a crucial role in the photocatalytic activity. Chaudhary et al. (2022) studied Cs_3_Bi_2_I_9_ nanodiscs (NDs) for photocatalytic aqueous HI splitting, as well as the photoelectrochemical conversion [71]. The authors were successful in stabilizing the NDs at various HI by maintaining a dynamic equilibrium between the saturated perovskite solution and the material precipitated. The perovskite NDs exhibited exceptional stability at a diluted concentration of HI for about 8 h. The synthesized nanostructures exhibited a remarkable H_2_ evolution rate of 22.5 μmol h^−1^ under visible light in an aqueous HI with apparent quantum efficiency of 1.3%. Furthermore, these NDs demonstrate a low overpotential of 533 mV at −100 mA per square centimeter for electrocatalytic H_2_ evolution in the same 6.34 M HI solution, with a turnover frequency of 11.7 H_2_ molecules per second. Notably, the NDs display exceptional recyclability and durability for both photocatalytic and electrocatalytic HI splitting processes [71]. Li et al. (2022) studied the anisotropic charge transfer of bismuth perovskite and indicated the favorable facets for the effective charge separation and enhanced photocatalytic hydrogen generation [72]. While (100) and (006) exhibited higher photocatalytic activity, the researchers developed a simple solvothermal process to achieve defined morphologies and facet-specific Cs_3_Bi_2_I_9_ perovskites to capture these advantageous anisotropic properties. Perovskite hexagonal prisms showed superior photocatalytic H_2_ evolution performance, splitting HI in ethyl acetate (EA) reached 1504.5 μmol h^−1^g^−1^ of H_2_ evolution rate, which is 22.1 times that of a disordered Cs_3_Bi_2_I_9_ photocatalyst compared to their counterpart samples with disordered structures and low (100)/(006) ratios. The stability and morphology of the perovskite were maintained after 8 h of photocatalytic reaction, with trace iodine particles formed on the (100) facets because of iodide photo-oxidation [72]. Miodynska et al. (2023) attempted to understand the effect of the larger cation in Bi-based 0-D perovskite in terms of morphology, stability, and band edge variation [73]. As a series of Bi-perovskites, A_3_Bi_2_I_9_ (A = Cs, Rb, MA, FA) are investigated for this purpose and identified that, although the crystal structure and bandgap is not affected, morphology was greatly influenced by the A-type cation. Meanwhile Cs forms hexagonal prisms, Rb forms irregular aggregates, and MA and FA form irregular structures of aggregated particles with a bandgap varying between 1.82 and 1.88 eV. Owing to the higher degree of crystallinity and defined morphology, Cs_3_Bi_2_I_9_ exhibited superior photocatalytic performance in both methanolic and acidic electrolytes: the Cs_3_Bi_2_I_9_ photocatalyst yielded approximately 35.5 μmol g^−1^ and 2304 μmol g^−1^ in the reaction when employed with 10% MeOH and HI/H_3_PO_2_ electrolytes, respectively, under 4 h of illumination [73]. 

Recently, Sb-based 0-D perovskite turned out to be an attractive option for photovoltaic and photocatalytic applications owing to its stability and non-toxic nature. Particularly, DFT studies of MA_3_Sb_2_I_9_ perovskite reveal high charge mobility, direct bandgap, and good defect tolerance. This motivated researchers to exploit these material properties for hydrogen generation. Ahmad et al. (2023) synthesized stable MA_3_Sb_2_I_9_ via solvothermal reaction for the hydrogen generation from aqueous HI using H_3_PO_2_ as a scavenger [74]. The material produces 300 μmol g^−1^ h^−1^ of H_2_ under visible light irradiation. The addition of Pt as a co-catalyst enhances the H_2_ production rate up to 883 μmol g^−1^ h^−1^. The material exhibits high stability and maintains its activity after five cycles [74]. The same group (2023) reported Cs_3_Sb_2_I_9_ with a comparable performance as compared to the MA-Sb based perovskite. Cs_3_Sb_2_I_9_ showed hydrogen generation of 95.54 μmol g^−1^. Platinum was used as bi-catalyst and obtained results for Cs_3_Sb_2_I_9_/Pt (2 mg) showed excellent hydrogen generation of 804.54 μmol g^−1^. As anticipated, the material exhibited higher stability as compared to its MA-based counterpart, owing to its all-inorganic composition [75]. Recently Rokesh et al. (2023) reported 2-(aminomethyl pyridine)SbI_5_ perovskite inspired materials for photocatalytic and PEC hydrogen generation [76]. Two photocatalysts, AMPS-1 and AMPS-2, are essentially synthesized from two Sb precursors, iodide, and oxide, and their hydrogen evolution rates are compared. The band of AMPS-2 is slightly larger than AMPS-1, which could possibly be due to the presence of oxide impurities. However, these impurities have not significantly affected the photocatalytic performance of the material: while AMPS-1 produced about 106.7 μmol g^−1^ h^−1^ hydrogen, AMPS-2 yielded 96.3 μmol g^−1^ h^−1^ [76].

#### 6.2.2. Tin and Germanium HPs

While tin–halide perovskites established their prominence in replacing the lead perovskites in solar cell devices, their phase instability and low bandgaps remain detrimental for their potential utilization in photocatalytic applications. On the other hand, Ge, being from the same group as Pb, is looked at as a prospective replacement for Pb in conventional perovskites. However, as far as we know, very little literature is available for conventional 3-D tin and Ge perovskites used in hydrogen generation. Fortunately, like that of lead perovskite, the electronic structure of tin perovskites is strongly influenced by organic cations and halogen, which can be exploited to tune their electronic energy levels and photo reactivity. Based on this strategy, Ricciarelli et al. (2022) suggested modifying conversional 3-D tin perovskite by replacing the MA cation with DMA, resulting in a stable perovskite with a suitable bandgap and energy level alignment for hydrogen generation [78]. Furthermore, PEA_2_SnBr_4_, DMASnBr_3_, and PhBz_2_GeBr_4_ showed remarkable stability in water and are employed in conjunction with g-C_3_N_4_ as heterojunction photocatalysts for water splitting [79,80,81]. These are discussed in heterojunction photocatalysts with g-C_3_N_4_ section.

#### 6.2.3. Vacancy-Ordered HPs

Another class of perovskites, vacancy-ordered (double) perovskites, have recently been investigated in various optoelectronic, electronic as well as energy conversion systems owing to their robust crystal structure and optoelectronic properties on par with conventional lead perovskites. Particularly, Sn-based perovskite Cs_2_SnX_6_, and Bi–Ag based perovskites Cs_2_AgBiX_6_ have proven to be suitable substitutes for the Pb-HPs for hydrogen generation.

Zhou et al. (2021) synthesized all-inorganic Cs_2_SnI_6_ perovskite material that can anchor single-atom platinum (Pt) sites with a unique Pt-I_3_ configuration on its surface (referred to as PtSA/Cs_2_SnI_6_), enhancing the photocatalytic activity and stability of the hydrogen evolution reaction. The Pt single-atom sites on the perovskite surface are decorated by using hydrothermal treatment, impregnation, and activation techniques. The PtSA/Cs_2_SnI_6_ catalyst has exceptional photocatalytic performance in hydrogen production from aqueous HI solution, achieving a remarkable turnover frequency of 70.6 h^−1^ per Pt. This value is approximately 176.5 times higher than that seen for Pt nanoparticles supported on Cs_2_SnI_6_ perovskite. Here is a summary of the photocatalytic test results. This superior performance of PtSA/Cs_2_SnI_6_ is attributed to the unique Pt–I_3_ configuration and the strong metal-support interaction effect, which enhanced the charge separation and transfer, and reduced the energy barrier for hydrogen production [77]. 

Recently, Pt and Pt group transition metal 0D vacancy-ordered halide perovskites have emerged as a notable class of materials for water splitting. Their exceptional stability, even when directly exposed to basic and acidic solutions without additional protection, has been highlighted. More details about this type of materials and the mechanisms for their extraordinary stability can be found in the recent review article [7]. 

### 6.3. Pb-Free HPs for Water Splitting via PEC Systems

Table 2 summarizes the Pb-free HP-based photoelectrode systems used in PEC for hydrogen generation. Often, 3-D perovskites (ABX_3_) suffer from instability issues in aqueous solutions and are usually employed with multiple protection layers to avoid direct contact with electrolytic mediums, particularly in PCE water splitting systems. Usually employed protection layers are InBiSn alloy, metal foils, including titanium and nickel, carbon materials, and atomic layer-deposited oxides. On the other hand, vacancy-ordered Pt, Ag–Bi, and Re-based perovskites exhibited resistance to degradation in harsh aqueous mediums, which enabled the use of perovskite electrodes without protection layers, opening new doors to PEC water splitting in variable pH mediums.

Cs_2_PtX_6_ vacancy-ordered perovskite is known for its exceptional stability over a wide range of pH, variable thermal and prolonged ambient exposure conditions [7]. Hamdan et al. (2020), studied the electrochemistry of Cs_2_PtI_6_ in electrolytic solution of pH varying from 1 to 14. Cyclic voltammetry (CV) is conducted to investigate the redox processes and catalytic activity of the materials in aqueous solutions of different pH values. The key finding was that Cs_2_PtI_6_ showed reversible oxidation/reduction peaks corresponding to I^−^/I^3−^ and Pt^4+^/Pt^2+^, and that it was stable in acidic, neutral, and basic media. Interestingly, Cs_2_PtI_6_ had a mixed valency of Pt^4+^ and Pt^2+^, and that the redox processes involved the formation and reduction of triiodide (I^3−^). Furthermore, Cs_2_PtI_6_ exhibited a photocurrent of 0.8 mA cm^−2^ at 1.23 V (vs. RHE) and over 12 h of PEC stability without loss of performance. However, this value is much lower than the reported values for lead halide perovskites which is possibly due to the low charge carrier mobility and lifetime of Cs_2_PtI_6_. The Cs_2_PtI_6_ material faces multiple challenges: its high production cost due to platinum usage limits its commercial viability; its 1.4 eV bandgap is not optimal for single-junction photovoltaics, but this could be improved through halide/chalcogen substitution in the X anion; and its poor contact with conducting glass leads to high series resistance and a low fill factor, indicating possible future work on optimizing deposition methods and charge transport layers [82]. Sikarwar et al. (2023) explored stable Ag–M (M = In, Bi, Sb) vacancy-ordered perovskite photoelectrodes for PEC water splitting. The synthesized materials exhibit remarkable resistance to oxidation, retaining structural stability over 100 cycles of electrochemical cycling. This allows for the utilization of these substances in the process of photoelectrochemical (PEC) water oxidation in both CH_3_CN and H_2_O, with and without the presence of an IrO_x_ cocatalyst [85]. 

Nandigana et al., (2023) studied two-dimensional variants of bismuth perovskite vacancy-ordered 2-D Cs_2_AgBiCl_6_ and 0-D Cs_3_Bi_2_Cl_9_ [84]. The hydrothermally synthesized double perovskite Cs_2_AgBiCl_6_ exhibits multifaceted crystal structures, featuring uniform di-pyramidal units, while Cs_3_Bi_2_Cl_9_ particles lack the defined structural formation. Interestingly both the perovskites exhibit high thermal stability, the TGA studies reveal that both the materials are stable up to 470 °C. The perovskite photoanodes are tested for the photoelectrochemical performance using three electrode electrochemical workstation in the 1 M KOH (pH∼13.7) alkaline electrolyte with Pt counter electrode and Ag/AgCl (Sat. KCl) as the counter electrode. The dark and light current densities of the Cs_2_AgBiCl_6_ are 3.30 mA cm^−2^ and 3.85 mA cm^−2^ and that of Cs_3_Bi_2_Cl_9_ are 1.78 mA cm^−2^ and 2.18 mA cm^−2^. The photo-response of both the photoanodes is instantaneous and persisted for about 300 sec. The PCE for the photoelectrochemical system is determined by the following equation and PCE of Cs_3_Bi_2_Cl_9_ and Cs_2_AgBiCl_6_ is achieved to be 0.09% and 0.13% at the 0.93 V vs. Ag/AgCl. The materials exhibited steady performance over 10 h. The low charge transfer resistance and high charge transfer density of Cs_2_AgBiCl_6_ as compared to the Cs_3_Bi_2_Cl_9_ resulted in its superior performance, which is understandably from the high crystallinity and defined morphology of the nanocrystals [84]. 

Chandra et al., (2023) the first time investigated for Re-based vacancy-ordered perovskite, Cs_2_ReX_6_ (X = Cl and Br) for PEC generation hydrogen [86]. The hydrothermally synthesized Re-perovskites have similar structure to Cs_2_PtX_6_ compounds and exhibited great thermal stability 600 °C. Furthermore, the Cs_2_ReX_6_ materials show panchromatic light absorption covering the entire visible region hinting high photon harnessing ability and with their valence band positions are below the water oxidation potential, they are suitable for photoanodes for solar water oxidation. Cs_2_ReX_6_ materials displayed their impressive electrochemical stability in a pH 11 solution, with no oxidation currents even at high potentials (up to 1 V vs. Ag/AgCl). The materials consistently yield photocurrent densities of 0.15–0.20 mA cm^−2^ at 0.4 V vs. Ag/AgCl under AM1.5 G illumination, remaining stable under both interrupted and continuous illumination. Also, Re-Br perovskite, due to its higher and suable flat-band potential and lower charge transfer resistance generated better photocurrent than Re-I perovskite [86]. 

Recently, Liu et al., (2023) studies the Cu-Ag based perovskite inspired material photoelectrodes for PEC water splitting [83]. The synthesized Cu_1.4_Ag_0.6_BiI_5_ nanocrystals exhibited a direct bandgap of 2.19 eV and band structures and well aligned with the redox potentials of water splitting reaction. To demonstrate the PEC from the material the photoanode is fabricated without an protective layer, and impressively the photoanode showed −0.05 V_RHE_ (RHE = reversible hydrogen electrode) onset potential and a photocurrent density of 4.62 mA cm^−2^ at 1.23 VRHE, as well as an applied bias photon-to-current efficiency (ABPE) of 2.94%. The charge recombination dynamics and transient absorption studies reveal the n-type nature of the material. Furthermore, the material displayed good stability in both solvent and water, which is attributed to the hydrophobic capping ligands involved during the synthesis process. The NCs-based photoanode for photoelectrochemical (PEC) water splitting exhibits a T50% lifetime of ~310 min, which is one of the longest reported lifetimes for lead-free perovskite-inspired materials [83]. 

## 7. Enhancing Photocatalytic Performance of Pb-Free HPs

Improving the photocatalytic performance and/or photocurrent of the electrode is crucial for ensuring sustainability of the lead-perovskite materials for hydrogen generation. Photocatalytic activity is highly dependent on the following: (1). ability to generate electron–hole pairs by harnessing maximum solar energy; (2). effective separation of the generated electron–hole pairs; (3). supporting redox reactions (on the surface in case of photocatalysis or photoelectrode immersed in the reaction solution. Approaches include (1). bandgap engineering by adapting suitable composition, and (2). forming heterojunctions, are essentially employed for improving the HER from the system.

Compositional engineering, on the other hand, is a promising tool to design stable perovskites. Careful choice of larger cation has a significant effect on the moisture-induced degradation of the perovskite. Molecules like DMA and PEA exhibit strong hydrophobicity, which protects the refrain the penetration of the water molecules into the perovskite lattice/structure. Unlike the usual Sn-perovskites, DMASnI_3_ exhibited exceptional stability. The material remained unaffected even after continuous exposure to the water for several hours.

### 7.1. Bandgap Tuning by Compositional Engineering

While the optical bandgap, absorption co-efficient and the nature of the bandgap are responsible for absorbing the photons and generating the charge carriers, tuning these inherent properties in the (lead-) halide perovskites is widely explored by developing mixed halide perovskite compositions and doping metal ions. In general, adjusting the bandgaps by varying the halide composition of the perovskite is the most explored method and, interestingly, mixed halide perovskites have superior optoelectronic properties, while bromide and chloride-based perovskites have long charge-carrier lifetimes, exceeding millisecond timescale, while iodide-based HPs have short charge-carrier lifetimes, not exceeding ns timescale however the absorption range is in the inverse order. Therefore, designing a mixed halide perovskites to balance out the properties based on the application is essential. For example, in lead HPs, creating funneling bandgaps by varying the halide composition creates a favorable energy band gradient for the effective charge transfer across the perovskite material [21]. Similarly, attempts are made to adjust the bandgap energy of CsPbX_3_ by anion exchange method resulting in the interfacial charge transfer and lifetime between the perovskite-TiO_2_ heterojunction. In addition to the bandgap tuning the studies revealed that chloride substitution reduces the trapping states, while iodide substitution leads to slower charge-carrier relaxation and indirect excitons with longer lifetime at the HPs/TiO_2_ interface [87]. This approach of tuning the bandgap and/or introducing the funneling bandgap to lead-free perovskites to improve photocatalytic activity has been studied by few researchers.

Tang et al. (2022) created a bandgap funneling effect for charge carriers within the bismuth-based double halide perovskites MA_3_Bi_2_Cl_9−*x*_I*_x_* by synthesizing the perovskite material via facile solvothermal anion-exchange technique [88]. The synthesized double halide perovskite exhibited a cubic crystal structure with lattice expansion, attributed to the I^−^ substitution. Furthermore, the EDS mapping showed the gradient distribution of I^−^ from the surface to the interior of the perovskites, indicating a bandgap funnel structure. Figure 7a shows the schematic representation of bandgap funneling effect in the MA_3_Bi_2_Cl_9−*x*_I*_x_*. The charge carried dynamics suggests a lower recombination rate and longer carrier lifetime. These eventually lead to higher photocurrent in mixed halide perovskites as compared to that of MA_3_Bi_2_Cl_9_. The photocatalytic performance of the perovskite samples was assessed for H_2_ production in a saturated HCl/HI solution (with H_3_PO_2_ as a reducing agent) under visible light irradiation (*λ* ≥ 420 nm, 100 mW cm^−2^), MA_3_Bi_2_Cl_8.8_I_0.2_ perovskite demonstrated highly efficient H_2_ evolution, yielding a rate of ≈214 µmol g^−1^ h^−1^. To further enhance H_2_ generation and inhibit the reverse reaction, 3.0 wt.% Pt cocatalyst was incorporated and the system exhibited enhanced photocatalytic activity, achieving a hydrogen evolution rate of ≈341 ± 61.7 µmol g^−1^ h^−1^ [88].

Chen et al. (2020) developed a novel lead-free perovskite material, Cs_3_Bi_2*x*_Sb_2−2*x*_I_9_ (CBSI-x), with potential applications as a photocatalyst for generating hydrogen from aqueous HI solutions [89]. The researchers investigated the effects of Sb doping on the crystal structure, electronic structure, optical absorption, charge transfer, and defect properties of Cs_3_Bi_2*x*_Sb_2−2*x*_I_9_ and the photocatalytic activity of the synthesized perovskite is compared to the conventional MAPI’s performance. The series of perovskites with variable Sb-dopant concentrations were synthesized through a co-precipitation method. The perovskite exhibited a hexagonal structure with space group P63/mmc with the peaks shifting to higher 2*θ* values with the increase of Sb in the perovskite. Figure 7b shows the absorption spectra of the Cs_3_Bi_2*x*_Sb_2−2*x*_I_9_ doped with variable concentration of Sb. While the doping of Sb in Cs_3_Bi_2*x*_Sb_2−2*x*_I_9_ effectively reduces the contribution of Bi^3+^ on the conduction band, this resulted in smaller bandgaps of the Bi–SB perovskites with stronger optical absorption than individual metal perovskites, particularly CBSI-0.3 posed the smallest bandgap (≈1.63 eV) and the smallest absorption tail. Moreover, the incorporation of Sb effectively reduced mid gap states, thereby leading to reduced non-radiative defects and improving charge transfer and separation mechanisms. Consequently, this material exhibited superior photocatalytic activity with enhanced photocurrent (when compared to the conventional lead-based perovskite) [89].

Yin et al. (2021), synthesized a series of stable and multifunctional Pt doped Sn-based perovskites Cs_2_Pt*_x_*Sn_1−*x*_Cl_6_ (0 < *x* < 1) via a simple hydrothermal method [90]. The perovskites demonstrated the switchable function for optoelectronics and photocatalysis applications. Figure 7c(i) and ii shows the crystal structure and transformation relationship of Cs_2_Pt*_x_*Sn_1−*x*_Cl_6_ and its charge-carrier dynamics model for *x* = 0.25 and 0.75. The perovskites have a face-centered cubic crystal structure with Fm$\bar{3}$m space group and form a series of continuous Cs_2_Pt*_x_*Sn_1−*x*_Cl_6_ solid solutions with isolated [SnCl_6_]^2−^ and [PtCl_6_]^2−^ octahedra. The study reveals that the Pt substitution not only enhances the thermodynamic stability and the thermal stability of Cs_2_Pt*_x_*Sn_1−*x*_Cl_6_ but also positively impacts the charge-carrier dynamics and the radiative recombination process of Cs_2_Pt*_x_*Sn_1−*x*_Cl_6_. With the Pt incorporation, the electron density surrounding chlorine improved the Pt-Cl bonding improving the stability of the crystal lattice. Additionally, the sub-bandgap states control the photoluminescence and photocatalytic functions of Cs_2_Pt*_x_*Sn_1−*x*_Cl_6_. The samples with low Pt content have higher sub-bandgap state-density and longer charge-carrier lifetime than those with high Pt content. In other words, as the Pt content increases, efficient self-trapping occurs, resulting in an enhanced radiative transition process. The substitution of Pt proves to be an effective method for adjusting the radiative recombination process and extended lifetime of photogenerated charge carriers. This has a direct correlation with the ultimate improvement of photocatalytic activity. The photocatalytic hydrogen evolution activity of Cs_2_Pt*_x_*Sn_1−*x*_Cl_6_ with triethanolamine (TEOA) as a sacrificial reagent under simulated solar light illumination. The hydrogen production rate followed an opposite trend of the photoluminescence intensity and quantum yield, indicating that different types of photogenerated charge carriers were involved in the two processes. This concludes that the hydrogen evolution activity to the sub-bandgap states in Cs_2_Pt*_x_*Sn_1−*x*_Cl_6_, which could capture or respond to the photoexcited carriers and facilitate the electron transfer process. The highest rate was 16.11 mmol g^−1^h^−1^ for Cs_2_Pt_0.05_Sn_0.95_Cl_6_ [90]. 

### 7.2. Pb-Free HP Heterojunctions

Effective charge separation by suppressing the recombination losses is one of the key factors to improve the photocatalytic (photoelectrochemical) activity of the hydrogen generation/production system. The electron–hole recombination is an inevitable phenomenon due to the Coulombic force between oppositely polar charges, moreover this swift process, with time limited to nanoseconds, is challenging to regulate. However, this can be minimized by engineering the photocatalyst and optimizing the system. Introducing heterojunctions is a classic and promising strategy to address the (surface) recombination in photocatalysts and is well explored in traditional solar-driven hydrogen-generation systems. Band alignment plays a crucial role in developing an appropriate heterojunction that ensures rapid charge separation, thereby improving the overall photocatalytic performance of the system. Lead-perovskite heterojunctions with TiO_2_, rGO, g-C_3_N_4_. etc., have been investigated and employed for photocatalytic activities, including hydrogen generation, CO reduction, and organic contaminants degradation [91,92,93,94,95,96,97]. These studies have inspired researchers to adapt lead-free perovskite-based heterojunction architectures for hydrogen generation.

#### 7.2.1. Semiconductor/Pb-Free Heterojunctions

Table 3 summarizes the semiconductor/Pb-free HP heterojunctions employed for the H_2_ generation. Jayaraman et al. (2021) investigated heterojunction of hydrothermally synthesized vacancy-ordered Pt-perovskites A_2_PtI_6_ (A = Cs^+^, Rb^+^, or K^+^) with BiVO_4_ for photoelectrochemical oxidation of water [98]. While synthesized Pt-perovskites exhibited impressive stability for about 2 weeks under ambient conditions, Cs_2_PtI_6_ showed higher thermal stability with decomposition temperature at ~364 °C. Furthermore, Cs_2_PtI_6_ exhibited remarkable stability in both acidic and alkaline environments, while Rb_2_PtI_6_ and K_2_PtI_6_ demonstrated instability. The heterojunction between BiVO_4_ and Cs_2_PtI_6_ led to enhanced photocurrent in comparison to BiVO_4_ alone, primarily attributed to improved charge separation and enhanced light absorption. Figure 8a shows the photocurrent response of BiVO_4_ and BiVO_4_/Cs_2_PtI_6_ heterojunction (Inset-Band alignment of the photoanode as a function of the wavelength). Furthermore, an IrO_x_ cocatalyst further boosted the photocurrent, achieving a level of 2 mA cm^−2^ at 1.23 V (vs RHE). However, the study is confined to the oxidation of water and can be potentially exploited for hydrogen generation for either acidic or basic reaction solutions [98]. 

Tang et al. (2020) demonstrated a novel perovskite/perovskite heterojunction catalyst by creating an interface between methylammonium bismuth iodide (MA_3_Bi_2_I_9_) and tri(dimethylammonium) hexa-iodobismuthate (DMA_3_BiI_6_) using a facile solvent engineering technique. This process lead to the formation of a series of heterojunction perovskites (BBP-0, BBP-1, BBO-5 and BBP10) through the reaction of MAI, HI, and Bi(NO_3_) in IPA with varying amounts of DMF [99]. The synthesis process and the formation of the MA_3_Bi_2_I_9_ and DMA_3_BiI_6_ are schematically represented as shown in Figure 8b. The VBM and CBM positions of MA_3_Bi_2_I_9_ and DMA_3_BiI_6_ showed a well-matched type-II heterostructure, which can facilitate the interfacial charge transfer between the two phases. It is suggested that photoexcitation results in electron transportation from the conduction band of DMA_3_BiI_6_ to that of MA_3_Bi_2_I_9_, while the holes on the valence bands migrate in the opposite direction. Thus, this perovskite–perovskite heterojunction enhances the charge separation and thereby improves the photocatalytic activity. The synthesized photocatalysts are employed for the H_2_ evolution from aqueous HI solution and identified that BBP-5 exhibited superior performance: BBP-5’s H_2_ production rate is 198.4 µmol g^−1^ h^−1^, which represents 15- and 8-fold increases, relative to BBP-0 and BBP-10, respectively [99]. 

Jiang et al. (2021) attempted to synthesize Cs_2_AgBiBr_6_ supported on nitrogen-doped carbon (N-C) for efficient photocatalytic hydrogen evolution from aqueous HBr solution under visible light irradiation [100]. Conventional one-pot synthesis is employed for the synthesis of the Cs_2_AgBiBr_6_/N-C) composited schematic representation of the synthesis process is shown in Figure 8c(i). The crystal structure of the perovskite is determined to be cubic and is not affected by the N-C interaction. The band alignment of the various heterojunction photocatalysts are shown in Figure 8c(ii): the CBM of Cs_2_AgBiBr_6_ is higher than the Fermi level of N-C, which enables photoexcited electron transfer from Cs_2_AgBiBr_6_ to N-C for subsequent H_2_ generation. The energy level difference is the largest for Cs_2_AgBiBr_6_/N-C-140, resulting in the strongest driving force of charge transfer. Furthermore, BET analysis reveals that N-C-140 has the highest surface area and nitrogen content, which facilitates electron diffusion and interfacial charge separation implicating that Cs_2_AgBiBr_6_/N-C-140 is expected to outperform. The photocatalytic activity of the photocatalyst is measured by the HER reaction from aqueous HBr solution. Cs_2_AgBiBr_6_/N-C-140 achieves a high hydrogen evolution rate of 380 μmol g^−1^ h^−1^, which is about 19 times faster than that of pure Cs_2_AgBiBr_6_ and 4 times faster than that of a physical mixture of Cs_2_AgBiBr_6_ and N-C-140. And the material exhibited a decent stability for six cycles with 3 h of hydrogen reaction per cycle [100]. 

Lin et al. (2023) studied a new 0-D perovskite halide, Cs_3_Rh_2_I_9_ (with dimer unit [Rh_2_I_9_]^3−^ separated by Cs ions) functionalized by nitrogen-doped carbon (NC) for hydrogen evolution for the chlorine–alkali electrolyte [101]. The Rb perovskite is prepared via solid state reaction using CsI as the flux followed by the dissolution–precipitation method to form Cs_3_Rh_2_I_9_ nanoclusters on a polar nitrogen-doped carbon (NC), as shown in Figure 8d(i). Cs_3_Rh_2_I_9_, due to their 0-D, experience downsizing to smaller clusters creating active sites of the NC due to repeated dissolution and precipitation process in aqueous DMF solution. The perovskite material undergoes self-electrochemical self-reduction and a bottom-up transformation, resulting in the formation of distinctive Rh nanoparticles. The fully reconstructed Cs_3_Rh_2_I_9_/NC-R catalyst significantly lowers the barrier for water dissociation in alkaline HER. The process is schematically represented in Figure 8d(ii,iii). Consequently, Cs_3_Rh_2_I_9_/NC-R exhibits an impressive mass activity of 772.1 mA mg^−1^Rh in a chlorine–alkali electrolyte. This figure is approximately 2.5 times greater than that of liquid-reduced Rh/NC catalysts with similar particle sizes and 35.5 times higher than electrochemically reduced Cs_3_Rh_2_I_9_-R catalysts with larger particle sizes [101]. 

#### 7.2.2. g-C_3_N_4_/Pb-Free HP Heterojunction

Carbon-based semiconductors, like reduced graphene or g-C_3_N_4_, have played a promising role in photocatalytic activities. Developing g-C_3_N_4_-based composite/heterojunction photocatalysts by anchoring the halide perovskite onto the g-C_3_N_4_ has an advantage of energy band alignment at the interfaces for enhancing the charge separation. Also, with surface functionalization of the g-C_3_N_4_ with amino or carboxyl groups, stronger chemical bond forms with the perovskite material rather than a mere physical adsorption, which further improves the charge injection at the interface. Moreover, g-C_3_N_4_, being a synthetic polymer, enables surface passivation of the perovskites and partially addresses the stability issues. In recent years, several researchers have reported several g-C_3_N_4_/Pb-free HP heterojunctions for hydrogen generation, as listed in Table 4.

Researchers have exploited the stability of 2-D perovskites to develop superior photocatalyst for hydrogen generation. However, 2-D perovskites are known for high bandgap and are to couple with suitable co-catalysts to facilitate the HER and in this case g-C_3_N_4_-based heterojunction is used to broaden the absorption spectra as well as support the HER. Romani et al. (2020) demonstrated PEA_2_SnBr_4_, a water-resistant 2D Pb-free HP, coupled with g-C_3_N_4_ co-catalyst system is employed for hydrogen photogeneration and organic dye degradation under visible light. The synthesized material retains its crystal structure and 2.67 eV bandgap even in contact with water for 4 h. The band alignment of the PES_2_SnBr_4_/g-C_3_N_4_ heterojunction is shown in Figure 9a. The hydrogen evolution rates (HERs) showed that the PEA_2_SnBr_4_/g-C_3_N_4_ composites had a remarkable enhancement of photocatalytic hydrogen production compared to pure g-C_3_N_4_ or PEA_2_SnBr_4_, with an optimal loading of 5 wt% perovskite. The composites also showed high HERs with glucose as a sacrificial agent. PEA_2_SnBr_4_/g-C_3_N_4_ composites exhibit synergistic effects, achieving high hydrogen evolution rates (1600 mmol g^−1^ h^−1^). This novel, water-resistant perovskite opens new possibilities for catalytic applications [80]. The group (2023) also introduced 2-D Ge-perovskites PhBz_2_GeBr_4_ (with a bandgap of 3.64 eV) and PhBz_2_GeI_4_ (with a bandgap of 3.32 eV) for the H_2_ production from aqueous triethanolamine (TEOA). The HER showed synergistic enhancement with heterojunction and Pt co-catalyst when compared to pure perovskites. For both PhBz_2_GeBr_4_ and PhBz_2_GeI_4_, the optimal perovskite loading was found to be 2.5 wt%, resulting in impressive hydrogen evolution rates of 81 mmol g^−1^ h^−1^ and 1200 mmol g^−1^ h^−1^, respectively. The iodine-based perovskite exhibited superior performance owing to its lower band and better band alignment with g-C_3_N_4_ which led to effective charge transfer and exhibited stability up to 4 cycles [81]. 

Bresolina et al. (2020) synthesized Cs_3_Bi_2_I_9_/g-C_3_N_4_ binary photocatalyst for hydrogen generation as well as degradation of organic compounds [102]. A simple processing by ultrasonication of Cs_3_Bi_2_I_9_ and g-C_3_N_4_ nanosheets resulted in nitrogen–iodine chemical interactions between the two semiconductors The estimated bandgaps of g-C_3_N_4_ and Cs_3_Bi_2_I_9_ were about 2.68 eV and 1.88 eV, respectively. The band alignment of Cs_3_Bi_2_I_9_/g-C_3_N_4_ is shown in Figure 9b. The pristine g-C_3_N_4_ sample exhibits solar-light photocatalytic activity through the hydrogen evolution reaction (HER) at a rate of 496.85 µmol g^−1^ h^−1^, attributed to the favorable bandgap and distinctive electronic structure of g-C_3_N_4_. Upon integration with metal halide perovskite, the photocatalytic performance of the heterojunction demonstrates an enhancement of approximately 46%, yielding a HER rate of approximately 920.76 µmol g^−1^ h^−1^ [102].

Wang et al. (2020) combined Cs_2_AgBiBr_6_ (CABB) with rGO by synthesizing the perovskite solid-state reaction followed by photo-reduction [103]. The SEM image of the synthesized photocatalyst is shown in Figure 9c(i)-inset. The schematic representation of the mechanism involved in the photocatalytic H_2_ evolution from aqueous HBr via CABB/2.5%RGO is shown in Figure 9c(i). In the saturated solution, CABB perovskite on the surface of RGO undergoes a dynamic precipitation–dissolution equilibrium process and RGO could act as an attachment for CABB particles. Under illumination, CABB generates charge carries and the generated electrons are transferred to conductive RGO through the M–O–C bonds and reduce H^+^ to produce H_2_ at the active sites of rGO. Meanwhile Br^−^ was oxidized to Br^3−^ by the holes on CABB particles. For CABB/2.5% rGO, the H_2_ evolution was observed to be 80 times greater than that achieved with pure CABB when both were immersed in a saturated HBr:H_3_PO_2_ solution uninterruptedly for 120 h [103]. 

Romani et al. (2020) introduced water stable DMASnBr_3_ as a co-catalyst for g-C_3_N_4_ for hydrogen evolution from 10% *v*/*v* aqueous triethanolamine (TEOA) reaction medium with the glucose sacrificial agent. Figure 9d shows the band alignment of the synthesized photocatalyst and the crystal structures of the DMASnBr_3_ and g-C_3_N_4_. Owing to the hydrophobic nature of the dimethylammonium cation, heterojunction showed an impressive stability of 4 h when dispersed in water. The composite showed an impressive HER of >1700 μmol g^−1^ h^−1^ in 10% (*v*/*v*) aqueous triethanolamine (TEOA) solution without any metal co-catalyst, under simulated solar light irradiation. This is a nearly 10-fold improvement compared to pure g-C_3_N_4_ and a 100-fold improvement compared to pure DMASnBr_3_. The composite also showed a high HER of 300 μmol g^−1^ h^−1^ in 0.1 M aqueous glucose solution, with 3 wt% Pt as co-catalyst [66]. The same group adapted a similar scheme for the synthesis of stable Bi-based perovskite and fabrication of its heterojunctions with g-C_3_H_4_ and demonstrated their application in photocatalytic hydrogen generation [81,103,104]. Medina-Llamas et al. (2023) employed simple and scalable methods, such as wet ball milling and thermal exfoliation to produce nanocrystalline Cs_3_Bi_2_Br_9_ and g-C_3_N_4_ nanosheets, respectively. The composites showed significant improvement of H_2_ production compared to the pure components. The highest HER is achieved at low perovskite loading (0.02 wt.%), while higher loadings result in a detrimental effect due to self-trapping phenomena in Cs_3_Bi_2_Br_9_. Also, this work highlights the importance of the use of photocatalysts in the nanocrystals and nanosheets, which enhanced the photocatalytic by improving the surface area [104]. 

Song et al. (2022) employed an in situ self-assembly method to yield CABB/g-C_3_N_4_ composite, which resulted in a type II heterojunction structure [105]. The band alignment of the synthesized composited is shown in Figure 9e. The optimal CABB/g-C_3_N_4_ composite exhibited an impressive hydrogen evolution rate of 60 μmol g^−1^ h^−1^, a notable 2.5-fold increase compared to pristine Cs_2_AgBiBr_6_ [105].

## 8. Prospectives and Conclusions

Lead-free perovskites turned out to be attractive active materials for solar-driven hydrogen generation owing to their excellent and tunable optoelectrical properties, ease of synthesis, and thin-film fabrication. In this review, the status of lead-free perovskite for hydrogen generation using photocatalytic and photoelectrochemical systems and state-of-the-art strategies to improve the performance of the active material in terms of increasing the HER are discussed. Based on the literature available, we take the liberty of concluding and commenting on the following observations:Dimensionality and bandgap: In general, most of the non-lead metals (excluding Sn and Ge) tend to crystallize in low-dimensional perovskite structures. These 0-D and 2-D perovskites inherently exhibit higher bandgaps, making them suitable for water splitting applications.Stability: Unlike conventional lead perovskite with an MA cation, lead-free perovskites with a Cs cation are structurally stable and allow crystallization of materials into its low-dimensional perovskite phases. All inorganic 0-D Bi (/Sb) perovskite and vacancy-ordered Sn, Ag –Bi etc., exhibited excellent stability in a water medium for several hours, proving their potential application in photocatalytic systems. Alternatively, polymer encapsulation, hydrophobic ligand-assisted nanoparticle stabilization and core-shell perovskites also enhance stability and can be exploited for photocatalytic water splitting.Co-catalysts: Loading Pt co-catalyst to improve the HER has become trivial, however this modification in the system improved hydrogen evolution drastically. Considering the total system cost, it is essential to explore alternatives to Pt. While halide perovskite/Pt-based photocatalytic systems are extensively studied, several other metal (Ni, Cu, Mn) and oxide (-perovskite) co-catalysts, in conjunction with lead-free perovskites, can be explored.Lead-free perovskites are usually employed in photocatalytic systems rather than in photoelectrochemical water splitting. Most possible reasons would be the challenges in formation of the uniform thin films for the fabrication of the photoanode due to their low dimensionality.Owing to their wide gap, several lead-free perovskite compositions can be excellent choices for coupling with Si cells to develop tandem photoanodes for photoelectrochemical water splitting.

Overall, lead-free halide perovskite water splitting systems need further developments to be considered effectively as sustainable technology for hydrogen generation. With the ongoing, dynamic, and persistent research on the lead-free perovskite material study, innovative approaches to address the stability and improve the performance of these materials have a great scope in water splitting or solar-driven energy conversion applications altogether.

## Figures and Tables

**Figure 1 nanomaterials-14-01914-f001:**
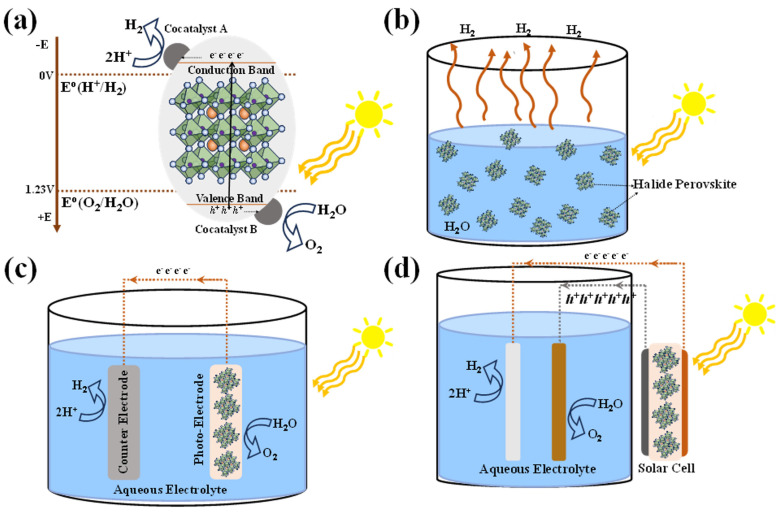
HP-based water splitting process and systems. (**a**) The process of photocatalytic water splitting employing two cocatalysts, A and B-cocatalyst. A is responsible for the H_2_O oxidation into O_2_ and cocatalyst B is responsible for the hydrogen evolution from the reduction of the H^+^ in the oxidation process. (**b**) Photocatalysis system, showing the dispersed photocatalyst in an aqueous medium. (**c**) Photo-electrocatalytic system, showing the perovskite-based photoelectrode, counter and reference electrodes immersed in the aqueous medium. (**d**) Photovoltaic-electrocatalytic system, showing the (perovskite-based) solar cell electrically connected to aqueous medium through the electrodes immersed in the medium.

**Figure 2 nanomaterials-14-01914-f002:**
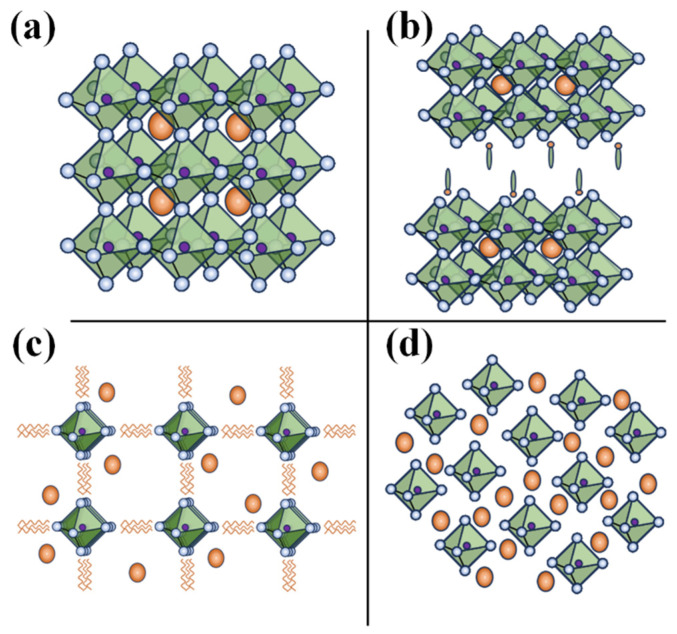
Structure of different dimensional perovskites: (**a**) 3-D; (**b**) 2-D; (**c**) 1-D; and (**d**) 0-D.

**Figure 3 nanomaterials-14-01914-f003:**
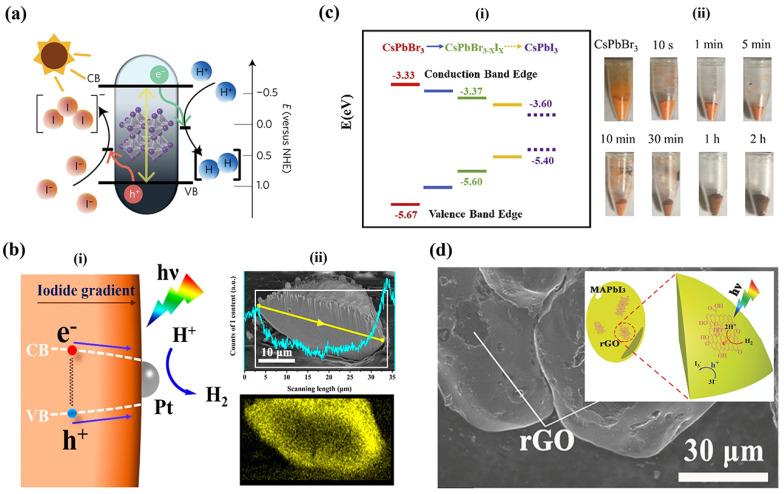
Photocatalytic Water Splitting using Pb-HPs: (**a**) Band alignment of MAPbI_3_ with respect to NHE. Reproduced with permission: Copyright 2016, Springer Nature Limited [6]. (**b**) (**i**). Bandgap funnel structure of MAPbBr_3−*x*_I*_x_*, with the bandgap narrowing to the surface, with Pt co-catalyst loaded on the surface of the perovskite. (**ii**). Cross-sectional SEM image MAPbBr_3−*x*_I*_x_* particle with the corresponding EDX mapping along the yellow line profile with I-element distribution profile. Reproduced with permission: Copyright 2018, American Chemical Society [21]. (**c**) (**i**). Band diagram of the CsPbBr_3_, CsPbBr_3−*x*_I*_x_* and CsPbI_3_. (**ii**). Photographic images of CsPbBr_3−*x*_I*_x_* powders with different halide exchange reaction times. Reproduced with permission: Copyright 2019, Elsevier [22]. (**d**) SEM image of MAPbI_3_/rGO (inset-Schematic illustration MAPbI_3_/rGO composite photocatalyst. Reproduced with permission: 2018 WILEY-VCH [23].

**Figure 4 nanomaterials-14-01914-f004:**
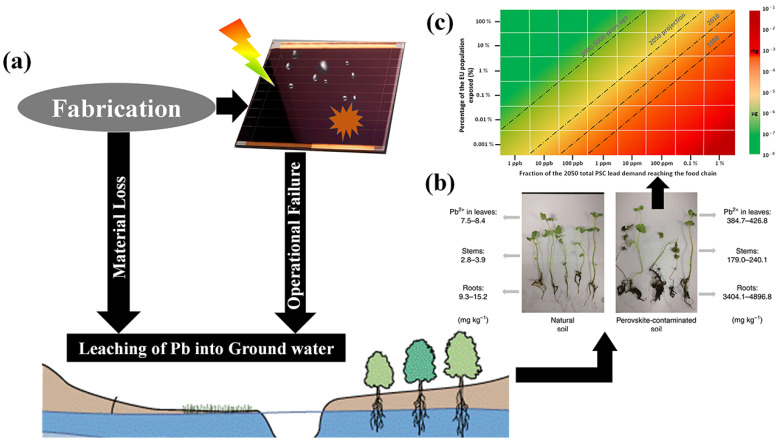
Toxicity of Pb-HP devices: (**a**) Illustration of possible leaching of lead into ground water. (**b**) The photographs of mint plants grown on control soil (**left**) and 250 mg kg^−1^ Pb^2+^ perovskite-contaminated soil (**right**). Reproduced with permission: Copyright 2020, Springer Nature [34]. (**c**) Assessment of the lead contamination on the human Lead Weekly intake level, considering the data for the world population and the total PSC lead necessary for electricity generation in 2050 vs. the adult Lead Weekly intake limit in 2010 and 3000 to 5000 years ago. Reproduced with permission: Copyright 2022, Elsevier Ltd. [35].

**Figure 5 nanomaterials-14-01914-f005:**
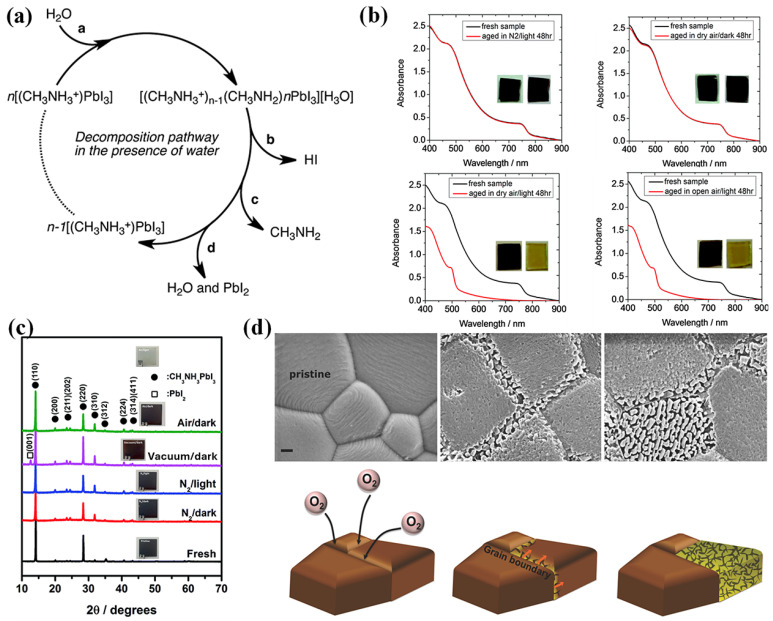
Stability of CH_3_NH_3_PbI_3_ Perovskite: (**a**) Decomposition of path of the perovskite in presence of water molecules A water molecule H_2_O is required to initiate the process with the decomposition, leading to the formation of hydrated MAPI and eventually to Pb and PbI_2_ with the gases HI and CH_3_NH_2_ leaving the substrate. Reproduced with permission: Copyright 2014, American Chemical Society [42]. (**b**) absorption spectra for the perovskite films on glass before (black) and after (red) aging in different conditions. The inset shows the photographs of films before (**left**) and after (**right**) aging. Reproduced with permission: Copyright 2016, The Royal Society of Chemistry [46]. (**c**) X-ray diffractograms of the perovskite films degraded in different atmospheres for 24 h. (The inset shows the photographs of films). Reproduced with permission: Copyright 2016, The Royal Society of Chemistry [47]. (**d**) SEM pictures of perovskite on ITO: pristine layers, initial stages of degradation, and later stages of degradation. The scale bars represent 200 nm followed by the illustration of the progression of oxygen induced degradation from the grain boundaries. Reproduced with permission: Copyright 2017, WILEY-VCH [48].

**Figure 6 nanomaterials-14-01914-f006:**
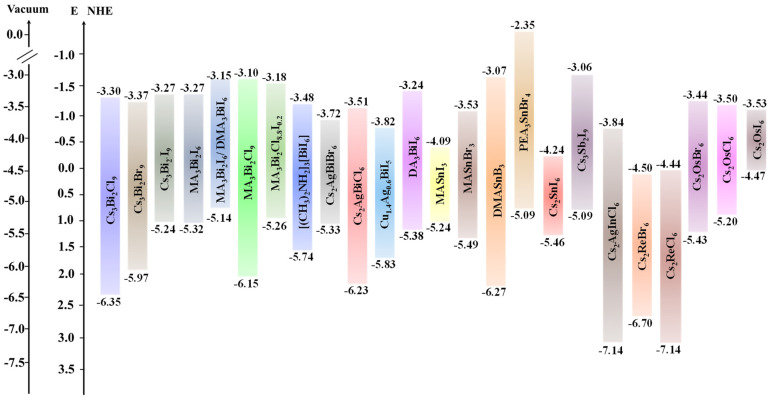
Bandgaps of various lead-free HP compositions discussed in this section.

**Figure 7 nanomaterials-14-01914-f007:**
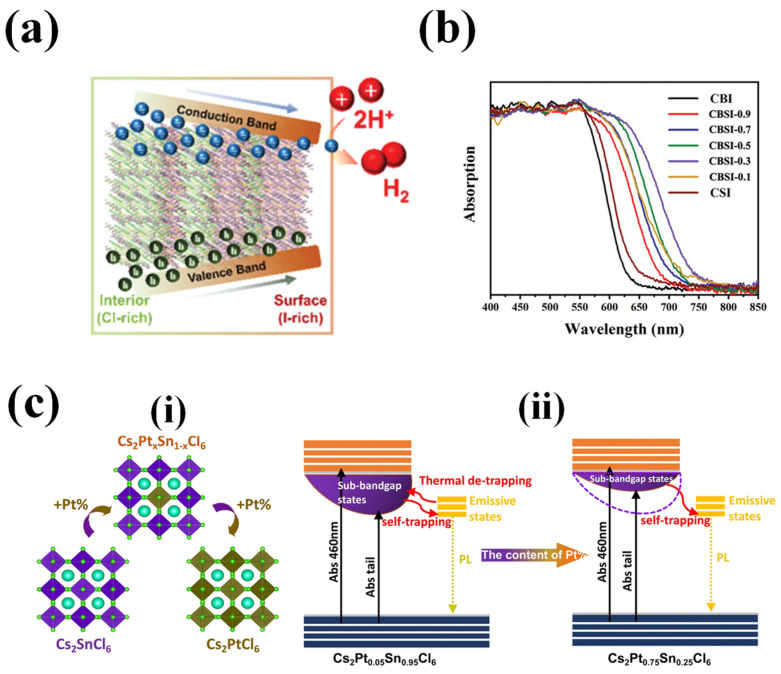
Compositional Engineering: (**a**) Schematic of Interfacial interaction of MA_3_Bi_2_Cl_9−*x*_I*_x_* with the bandgap funnel structure. Reproduced with permission: Copyright 2022, Wiley-VCH [88]. (**b**) Absorption spectra of the as-prepared Cs_3_Bi_2*x*_Sb_2−2*x*_I_9_ (*x* = 0.9, 0.7, 0.5, 0.3, 0.1). Reproduced with permission: Copyright 2020, Wiley-VCH [89]. (**c**) (**i**). Crystal structure and transformation relationship of Cs_2_Pt*_x_*Sn_1−*x*_Cl_6_. (**ii**). Charge-carrier dynamics model of Cs_2_Pt_0.05_Sn_0.95_Cl_6_ and Cs_2_Pt_0.75_Sn_0.25_Cl_6_. Reproduced with permission: Copyright 2021, Wiley-VCH [90].

**Figure 8 nanomaterials-14-01914-f008:**
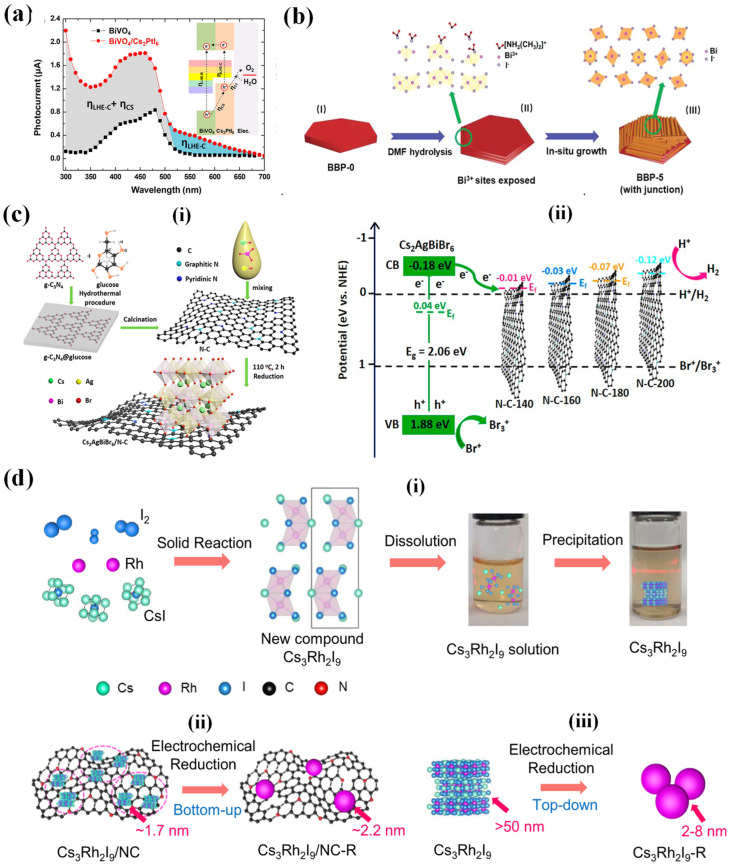
Semiconductor/Pb-free HP heterojunctions: (**a**) Photocurrent response of BiVO_4_ and BiVO_4_/Cs_2_PtI_6_ heterojunction (Inset-Band alignment of the photoanode as a function of the wavelength). Reproduced with permission: Copyright 2021, American Chemical Society [98]. (**b**) Schematic representation of the formation of MA_3_Bi_2_I_9_/DMA_3_BiI_6_ heterojunctions. (I) MA_3_Bi_2_I_9_ in the pre-reaction solution prior to the solvothermal process. (II) The DMA^+^ ions generated through DMF hydrolysis reacted with the unreacted bismuth ions on the MA_3_Bi_2_I_9_ perovskites. (III) In-situ formation of MA_3_Bi_2_I_9_/DMA_3_BiI_6_ perovskite heterojunctions in BBP-5. Reproduced with permission: Copyright 2020, Wiley-VCH [99]. (**c**) Schematic representation of synthesis of Cs_2_AgBiBr_6_/N-C Photocatalyst. (**i**) Conventional one-pot synthesis is employed for the synthesis of the Cs_2_AgBiBr_6_/N-C). (**ii**) The band alignment of the various heterojunction photocatalysts.Reproduced with permission: Copyright 2021, American Chemical Society [100]. (**d**) (**i**). Schematic representation of Cs_3_Rh_2_I_9_ bulk crystal. (**ii**). Electrochemical reduction of Cs_3_Rh_2_I_9_ clusters on NC (Cs_3_Rh_2_I_9_/NC) to form Cs_3_Rh_2_I_9_/NC-R. (**iii**). Electrochemical reduction of bulk Cs_3_Rh_2_I_9_ to form Cs_3_Rh_2_I_9_-R with large particle size. Reproduced with permission: Copyright 2023, Springer Nature [101].

**Figure 9 nanomaterials-14-01914-f009:**
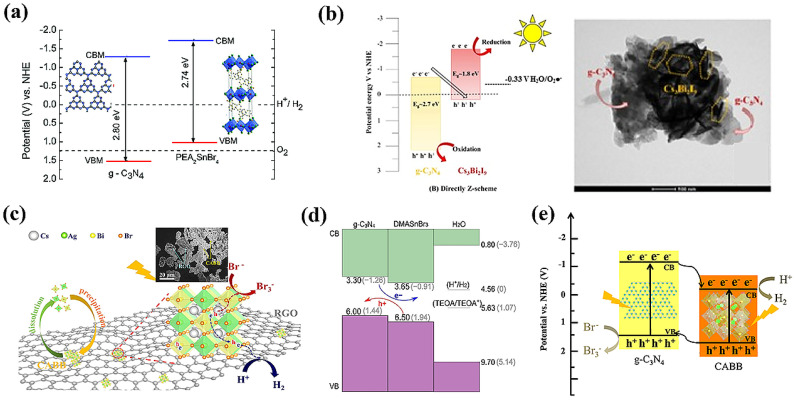
g-C_3_N_4_/Pb-free HP Heterojunctions: (**a**) Band alignment of the PEA_2_SnBr_4_ and g-C_3_H_4_ relative to NHE potential. Reproduced with permission: Copyright 2020, Royal Society of Chemistry [80]. (**b**) Band alignment of the Cs_3_Bi_2_I_9_ and g-C_3_H_4_ relative to NHE potential. Reproduced with permission: Copyright 2020, Elsevier B.V. [102]. (**c**) Schematic representation of H_2_ evolution mechanism by Cs_2_AgBiBr_6_-rGO under visible light irradiation (inset: SEM image of the photocatalyst). Reproduced with permission: Copyright 2019, Elsevier B.V. [103]. (**d**) Band alignment of g-C_3_N_4_ and DMASnBr_3_ aligned with respect to water, H^+^/H_2_ and TEOA/TEOA^+^ redox level. Reproduced with permission: Copyright 2020, Wiley-VCH [79]. (**e**) Band alignment of the Cs_2_AgBiBr_6_ and g-C_3_H_4_ relative to NHE potential. Reproduced with permission: Copyright 2020, Springer Nature [105].

**Table 1 nanomaterials-14-01914-t001:** 0-D bismuth and antimony HP photocatalytic systems discussed in Section 6.3.

No	Material	Reaction Medium	Illumination	HER/Photocurrent	Stability	Ref
1	MA_3_Bi_2_I_9_/Pt	Aqueous HI/H_3_PO_2_	300 W Xe-lamp with a 400 nm cutoff filter	169.21 µmol g^−1^ h^−1^	10 h of 7 cycles	[70]
2	Cs_3_Bi_2_I_9_	Aqueous HI/H_3_PO_2_	Visible Light	22.5 μmol h^−1^ 11.7 H_2_ molecules per second	5 h of 3 cycles	[71]
3	Cs_3_Bi_2_I_9_	HI in n ethyl acetate	100 mW cm^−2^	1504.5 μmol g^−1^ h^−1^	2 h of 4 cycles	[72]
4	Cs_3_Bi_2_I_9_/Pt	Aqueous HI/H_3_PO_2_ Aqueous MeOH	100 mW cm^−2^ (λ > 420 nm)	2304 μmol g^−1^ 35.5 μmol g^−1^	4 hrs	[73]
5	MA_3_Sb_2_I_9_/Pt	Aqueous HI/H_3_PO_2_	100 mW cm^−2^ (λ > 400 nm)	883	3 h of 4 cycles	[74]
6	Cs_3_Sb_2_I_9_/Pt	Aqueous HI/H_3_PO_2_	100 mW cm^−2^ (λ ≥ 400 nm)	804.54 μmol g^−1^	50 h	[75]
7	2-AMPSbI_5_-1	Sodium sulfate: H_2_O	150 W xenon lamp	106.7	4 cycles	[76]
8	2-AMPSbI_5_-2	Sodium sulfate: H_2_O	150 W xenon lamp	96.3	4 cycles	[76]
9	PtSA/Cs_2_SnI_6_	Aqueous HI	100 mW cm^−2^ (λ ≥ 420 nm)	430 μmol g^−1^ h^−1^	180 h	[77]

**Table 2 nanomaterials-14-01914-t002:** List of Pb-free HPs for PEC systems for hydrogen generation discussed in Section 6.3.

No.	Material	Photoanode Area	Electrolyte/Illumination	Photocurrent	Stability	Ref
1	Cs_2_PtI_6_		pH-11 1 sun (AM 1.5 G, 100 mW cm^−2^)	0.8 mA cm-^2^at 1.23 V	12 h	[82]
2	Cu_1.4_Ag_0.6_BiI_5_	0.785 cm^2^	1 sun (AM 1.5 G, 100 mW cm^−2^)	4.62 mA cm^−2^ at 1.23 V_RHE_	~5 h	[83]
3	Cs_2_AgBiCl_6_	1 cm^2^	1 M KOH 1 Sun	3.85 mA @ 1.0 V (vs. Ag/AgCl)	10 h	[84]
4	Cs_3_Bi_2_Cl_9_	1 cm^2^	1 M KOH 1 Sun	3.85 mA @ 1.0 V (vs. Ag/AgCl)	10 h	[84]
5	Cs_2_AgInCl_6_	-	water and acetonitrile	0.75 mA cm^−2^ @ 600 mV (vs. RHE)	2 h	[85]
6	Cs_2_ReBr_6_	25 mm^2^	1.5 mM KOH solution 1 Sun	0.20 mA cm^−2^ 0.4 V vs. Ag/AgCl	24 h	[86]
7	Cs_2_ReI_6_	25 mm^2^	1.5 mM KOH solution 1 Sun	0.14 mA cm^−2^ 0.4 V vs. Ag/AgCl	24 h	[86]

**Table 3 nanomaterials-14-01914-t003:** List of Semiconductor/Pb-free HP heterojunctions discussed in Section 7.2.1.

Heterojunction	Reaction Solution	Light Source	HER (µmol g^−1^ h^−1^)	Stability	Photocurrent	Ref
BiVO_4_/Cs_2_PtI_6_	H_2_O:KOH	500 Wm^−2^, AM 1.5G filter		-	2 mA cm^−2^ at 1.23 V (vs. RHE)	[98]
Cs_2_AgInCl_6_/IrO*_x_*	CH_3_CN:H_2_O	1 Sun		2 h	155.8 mA @ 600 mV (vs. RHE)	[85]
MA_3_Bi_2_I_9_/DMA_3_BiI_6_	H_2_O:HBr	100 mW cm^−2^ (λ ≥ 420 nm)	198.2	10 h/10 cycles		[99]
2-AMPSbI_5_/GO	sodium sulfate:H_2_O	150 W xenon lamp	185.8	4 cycles		[76]
Cs_2_AgBiBr_6_/N-C	H_2_O:HBr	λ ≥ 420 nm	380	3 h/6 cycles		[100]
Cs_3_Rh_2_I_9_/NC-R	H_2_O:KOH			50 h	mass activity of 772.1 mA mg^−1^ (10 mA cm^−2^ at 1.23 V (vs. RHE)	[101]

**Table 4 nanomaterials-14-01914-t004:** List of g-C_3_N_4_/Pb-free HP heterojunctions discussed in Section 7.2.2.

Material	Reaction Solution	Light Source	Hydrogen Evolution Rate (µmol g^−1^ h^−1^)	Stability	Ref
PEA_2_SnBr_4_	H_2_O/10% TEOA	500 Wm^−2^, AM 1.5G filter	1613	-	[80]
PhBz_2_GeI_4_	H_2_O/10% TEOA	500 Wm^−2^, AM 1.5G filter	1200	6 h/4 cycles	[81]
Cs_3_Bi_2_I_9_	H_2_O/10% MeOH	450 W Xe lamp	920.76	6 h	[102]
Cs_2_AgBiBr_6_-rGO	H_2_O/HBr	>420 nm	48.9	10 h/12 cycles	[103]
DMASnX_3_	H_2_O/10% TEOA	500 Wm^−2^, 300–800 nm	1730	4 h	[79]
Cs_3_Bi_2_Br_9_	H_2_O/10% TEOA	500 Wm^−2^, 300–800 nm	4593	-	[104]
Cs_2_AgBiBr_6_	HBr/20% H_3_PO_2_	300 W (λ ≥ 420 nm)	60	3 h/14 cycles	[105]

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
