# Peer review of "Prospects of Halide Perovskites for Solar-to-Hydrogen Production"

_nanomaterials, 2024, doi:10.3390/nano14231914_

Round 1
Reviewer 1 Report
Comments and Suggestions for Authors
The authors reviewed halide perovskite for solar-to-hydrogen production, unveiling the potential of lead-free halide perovskite. Overall, the review is well organized. The reviewer has some comments/suggestions:
1. The author listed the stability factors related to perovskite in section 4.2, like moisture, oxygen and light, but another important factor “Tolerance Factor” related to crystal structure has great influence on the stability of perovskite. Therefore, the reviewer recommends the authors to discuss about this factor.
2. In Table 2, how about the stability of number 6 and 7?
3. Are there any new development in literatures related to the topic of this review this year? I notice most articles cited were published before 2023. I recommend the authors add some latest reports, such as Nat Energy 9, 272–284 (2024); Angew. Chem. Int. Ed. 2024, e202411016; Chem. Eng. J. 492, 152024 (2024); Energy Environ. Sci., 2024,17, 3604-3617, etc.
4.The authors need to pay attention to some writing details, the following are some examples:
a. Page 3 line 109, all valuables should be in italics.
b. Page 9 line 332, spelling mistake of moisture.
c. Page 13 line 511, note the superscript format for the plus symbol.
d. In Table 4, there should be a space between number and unit.
e. Page 18 lines 712 and 728, there are two 5.3.2.
f. Page 19 line 747, 5.3. should be 5.4.
Reviewer 2 Report
Comments and Suggestions for Authors
In this review, Bansal and colleagues summarize recent advancements and future prospects of perovskite materials for hydrogen production, with a particular emphasis on lead-free perovskites. While there are numerous review articles on this topic, the additional take-home message from this work is unclear. Nonetheless, I appreciate the authors' efforts, which reflect their expertise in the field. Overall, the review is well-written. Given the authors' focus on lead-free perovskites, I recommend incorporating this keyword into the title. This adjustment could help readers to navigate through the crowded literature on perovskites as catalysts for solar hydrogen production. There are some additional minor issues that need to be addressed before the paper can be recommended for publication.
1. There are many occasions where the citations are missing. The authors must go through the manuscript carefully and correct that issue. Here are some examples out of those:
a. In the context of the second sentence in the Introduction, “Ever since the first demonstration of the photoelectrochemical water splitting by Honda and Fujishima, solar energy driven hydrogen generation by employing semiconductors as an active material created a buzz in the research community.” Citation missing.
b. “Wang and colleagues employed a similar strategy to enhance HI splitting by MAPbI3, using Pt/TiO2 nanoparticles as nanoscale electron-transporting channels to efficiently extract electrons when in contact with MAPI.” Citation missing.
c. “Ram et al. conducted calculations and predicted that the electricity energy demand from PVs in 2050 would be 12,210 TWh/year, accounting for 60% of the total electricity demand.” Citation missing.
2. In the section “Solar-Driven Hydrogen Generation Systems” while explaining the basics, the authors many times mentioned about the importance of band gap and band gap tunability. In my opinion, no doubt the band gap is important, but the energy level of the valence band and conduction band are crucial. Two materials with similar band gap does not necessarily means that they will show activity; their absolute VB and CB positions are crucial. Authors should emphasize this point.
3. It will be better flow for the readers if the subsection “Classification of HP perovskites (based on their structure)” is brought just before the section “Lead-Halide Perovskites for Hydrogen Generation”. Inducing the perovskites (Pb based/lead free) first and then talk about their implications makes more sense.
4. Some of the relevant review articles must be appreciated by including them as references. https://doi.org/10.1016/j.jechem.2020.08.057, https://doi.org/10.1016/j.ijhydene.2024.07.039
Reviewer 3 Report
Comments and Suggestions for Authors
This article systematically introduces the application status of Lead-Halide Perovskites for Hydrogen Generation, the problems existing in lead-containing perovskites, the specific characteristics of lead-free perovskites, and reviews their enhancement of photocatalytic performance. This content has certain guiding significance for further study of lead-free perovskite batteries suitable for photocatalytic systems.
Reviewer 4 Report
Comments and Suggestions for Authors
Prospects of Halide Perovskites for Solar-to-Hydrogen Production
This review comprehensively examines the current research on solar-to-hydrogen production utilizing perovskite materials, with a critical analysis of the limitations associated with traditionally used lead-containing perovskites. It also provides a thorough overview of the progress in lead-free perovskite alternatives, underscoring the importance of developing materials that prioritize environmental safety and enhanced stability. The manuscript holds substantial potential as a high-quality review article, contingent on the revision of a few specific elements for greater clarity and completeness.
1. The description accompanying Figure 1 categorizes the systems into three main types: photocatalytic, photoelectrochemical (PEC), and photovoltaic-powered electrolysis (PV-E). However, there actually are more variation in the device configuration. It would be beneficial to introduce more specified device configuration, such as tandem and monolithic devices, considering the idealized devices for water splitting (doi.org/10.1039/D3SE01371E).
2. The depiction of the MAPI decomposition process in water, as shown in Figure 4(a), lacks clarity. It would be preferable to utilize an alternative figure that more effectively illustrates the detrimental impact of moisture on lead halide perovskites and conveys the decomposition mechanism with greater detail. It is also recommended to include a simplified mechanistic equation, as outlined in the Oxygen and Photo-induced Degradation section below.
3. Recent examples of highly efficient perovskite-based solar-to-hydrogen conversion devices > 10% STH should also be cited. (doi.org/10.1002/smll.202300174; 10.1039/D0EE02959A)
4. The overall impact of this review paper appears somewhat limited. For example, the discussion on improving stability of Lead-Free Halide Perovskites for hydrogen production through dimensionality and bandgap engineering has been mentioned in other review papers as well (e.g., Lead-Free Metal Halide Perovskites for Hydrogen Evolution from Aqueous Solutions, Current Trends in Strategies to Improve Photocatalytic Performance of Perovskite Materials for Solar to Hydrogen Production). To enhance the impact of this paper, it would be desirable to provide a detailed explanation of the differences in degradation mechanisms caused by moisture, oxygen, and photo-induced factors between lead-based and lead-free halide perovskites, and to elucidate their underlying causes.
